# Genome-wide analysis identifies molecular systems and 149 genetic loci associated with income

W. David Hill[1,2]*, Neil M. Davies [3,4], Stuart J. Ritchie[5], Nathan G. Skene [6,7,8], Julien Bryois [9],
Steven Bell [10,11,12], Emanuele Di Angelantonio[10,11,12,13], David J. Roberts[14,15,16], Shen Xueyi [17], Gail Davies[1,2],
David C.M. Liewald [1,2], David J. Porteous [1,18], Caroline Hayward [19], Adam S. Butterworth [10,11,12],
Andrew M. McIntosh [1,17], Catharine R. Gale [1,2,20,21] & Ian J. Deary[1,2,21]

Socioeconomic position (SEP) is a multi-dimensional construct reflecting (and influencing) multiple socio-cultural, physical, and environmental factors. In a sample of 286,301 participants from UK Biobank, we identify 30 (29 previously unreported) independent-loci associated with income. Using a method to meta-analyze data from genetically-correlated traits, we identify an additional 120 income-associated loci. These loci show clear evidence of functionality, with transcriptional differences identified across multiple cortical tissues, and links to GABAergic and serotonergic neurotransmission. By combining our genome wide association study on income with data from eQTL studies and chromatin interactions, 24 genes are prioritized for follow up, 18 of which were previously associated with intelligence. We identify intelligence as one of the likely causal, partly-heritable phenotypes that might bridge the gap between molecular genetic inheritance and phenotypic consequence in terms of income differences. These results indicate that, in modern era Great Britain, genetic effects contribute towards some of the observed socioeconomic inequalities.

[1] Centre for Cognitive Ageing and Cognitive Epidemiology, University of Edinburgh, 7 George Square, Edinburgh EH8 9JZ, UK. [2] Department of Psychology, University of Edinburgh, 7 George Square, Edinburgh EH8 9JZ, UK. [3] Medical Research Council Integrative Epidemiology Unit, University of Bristol, Bristol BS8 2BN, UK. [4] Bristol Medical School, University of Bristol, Bristol BS8 2BN, UK. [5] Social, Genetic and Developmental Psychiatry Centre, King's College London, London, UK. [6] Laboratory of Molecular Neurobiology, Department of Medical Biochemistry and Biophysics, Karolinska Institutet, Stockholm, Sweden. [7] UCL Institute of Neurology, Queen Square, London, UK. [8] Department of Medicine, Division of Brain Sciences, Imperial College, London, UK. [9] Department of Medical Epidemiology and Biostatistics, Karolinska Institutet, Stockholm, Sweden. [10] The National Institute for Health Research Blood and Transplant Unit in Donor Health and Genomics at the University of Cambridge, University of Cambridge, Strangeways Research Laboratory, Wort's Causeway, Cambridge CB1 8RN, UK. [11] UK Medical Research Council/British Heart Foundation Cardiovascular Epidemiology Unit, Department of Public Health and Primary Care, University of Cambridge, Strangeways Research Laboratory, Wort's Causeway, Cambridge CB1 8RN, UK. [12] British Heart Foundation Centre of Excellence, Division of Cardiovascular Medicine, Addenbrooke's Hospital, Hills Road, Cambridge CB2 0QQ, UK. [13] NHS Blood and Transplant, Cambridge, UK. [14] Cambridge Substantive Site, Health Data Research UK, Wellcome Genome Campus, Hinxton, UK. [15] BRC Haematology Theme and Radcliffe Department of Medicine, University of Oxford, Oxford, UK. [16] NHS Blood and Transplant – Oxford Centre, Oxford, UK. [17] Division of Psychiatry, University of Edinburgh, Edinburgh EH10 5HF, UK. [18] Centre for Genomic and Experimental Medicine, Institute of Genetics and Molecular Medicine, University of Edinburgh, Western General Hospital, Edinburgh EH4 2XU, UK. [19] MRC Human Genetics Unit, MRC Institute of Genetics and Molecular Medicine, University of Edinburgh, Western General Hospital, Edinburgh EH4 2XU, UK. [20] MRC Lifecourse Epidemiology Unit, University of Southampton, Southampton SO16 6YD, UK. [21]These authors contributed equally: Catharine R. Gale, Ian J. Deary. *email: David.Hill@ed.ac.uk

People with advantaged socioeconomic backgrounds, on average, live longer, and have better mental and physical health than those from more deprived environments[1–3]. An understanding of the causes underlying the association between socioeconomic position (SEP) and health is likely to be helpful to minimize social disparities in health and well-being[4].

The link between SEP and health is typically thought to be due to environmental factors, including, but not limited to access to resources, exposure to harmful or stressful environments, adverse health behaviours, such as smoking, poor diet and excessive alcohol consumption, and a lack of physical exercise[5]. However, genetic factors (most likely via mediated pleiotropy, Fig. 1) have been discussed as a partial explanation for the SEP–health gradient; for example, genetic predispositions towards certain diseases and/or genetic influences on what foods people like, could lead to poor diet, which in turn could lead to both lower SEP and poorer health[6]. It has recently been demonstrated that genome-wide association studies (GWASs) can capture shared genetic associations between measures of health and SEP[7]. Potential pleiotropic effects are highlighted in the observed genetic correlations between SEP variables, such as years of education completed, household income and social deprivation, and physical and mental health traits including longevity[7,8].

Loci associated with two SEP phenotypes, education and household income, have been identified via GWASs[7,9–11], but—consistent with other complex traits, such as height—these loci collectively account for only a small fraction of the total heritability of the traits in question. For household income, an analysis of a sample of 96,900 individuals from Great Britain found that additive genetic effects tagged by common single-nucleotide polymorphisms (SNPs) accounted for ~11% (SE = 0.7%) of differences in household income[7]. Two loci attained genome-wide

significance in that study, but they collectively accounted for <0.005% of the total SNP heritability.

Here, we use the UK Biobank data set[12] to examine genetic associations with household income ($n = 286,301$) in a contemporary British sample. We identify 30 independent genome-wide significant loci, 29 of which are unreported in previous work. Using a method that leverages power from genetically correlated traits, multi-trait analysis of GWAS (MTAG), an additional 120 loci are found to be associated with income. We identify neurogenesis and the components of the synapse as being associated with income. Furthermore, we link transcription differences across multiple cortical tissue types, as well as both GABAergic and serotonergic neurotransmission, to income differences. We also show that the genes linked to differences in income are predominantly those that have been previously linked with intelligence[8], and that intelligence is one of the likely causal factors leading to differences in income. We compare the genetic correlations derived using income with those derived using another measure of SEP, educational attainment, to show that the genetic variants associated with income are related to better mental health than those related to education. Finally, we predict 2.5% of income differences using genetic data alone in an independent sample.

## Results

**Graphical representation of statistical analysis.** A flow chart summarizing all statistical analyses conducted is displayed in Fig. 2.

**SNP-based analysis of income.** For household income, 3712 SNPs attained genome-wide significance ($P < 5 \times 10^{-8}$), across 30 independent loci (Fig. 3a and Supplementary Data 1), which contained 68 independent significant SNPs and 31 lead SNPs. A

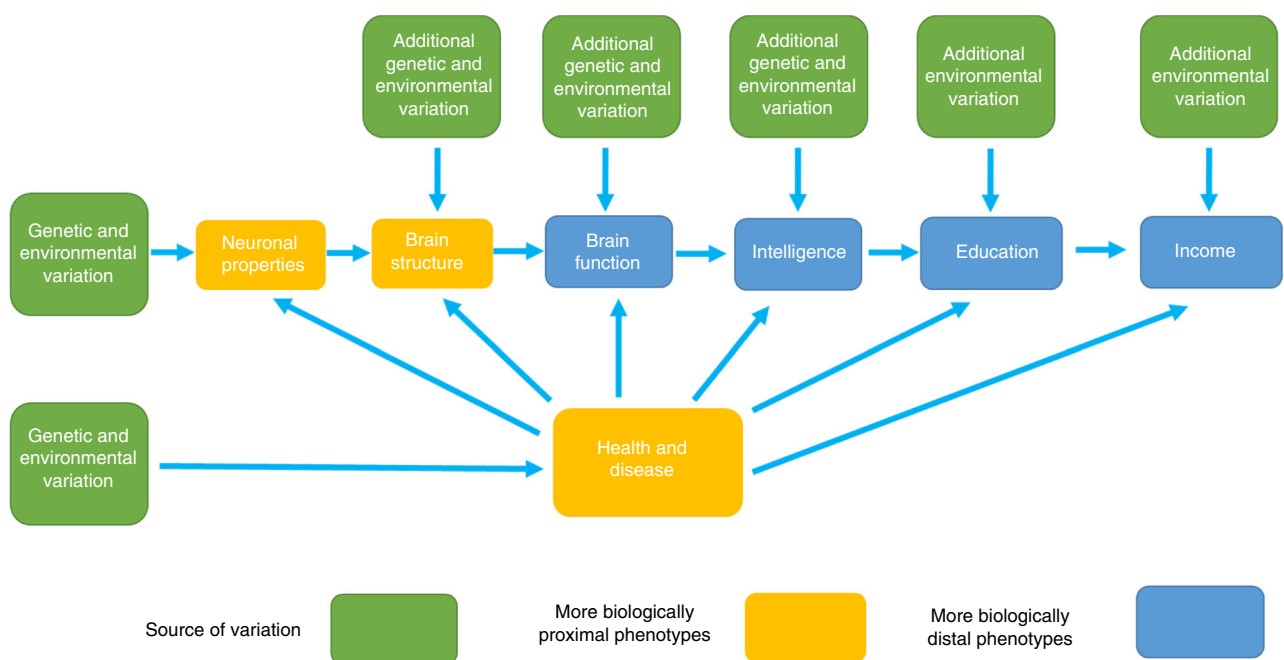

**Fig. 1 Illustrating a possible pathway from genetic inheritance to income.** In this pathway there are no direct effects of genetic variants on income. Rather, mediated pleiotropy (also termed vertical pleiotropy) is used to understand, in part, the link between genetic variation and more biologically distal phenotypes such as income and education. Mediated pleiotropy describes instances where genetic variation is linked to a phenotype (in this case income) through genetic effects that act on another partly heritable trait. These partly heritable traits would also be associated with income, and so the genetic effects that act on them would also be associated with income. For simplicity, this schematic illustrates only two possible pathways between genetic variation and income. In reality, there may be, and are likely to be, many links between genetic variation, including bidirectional causality between the phenotypes in the pathway, and the more biologically distal phenotypes such as income.

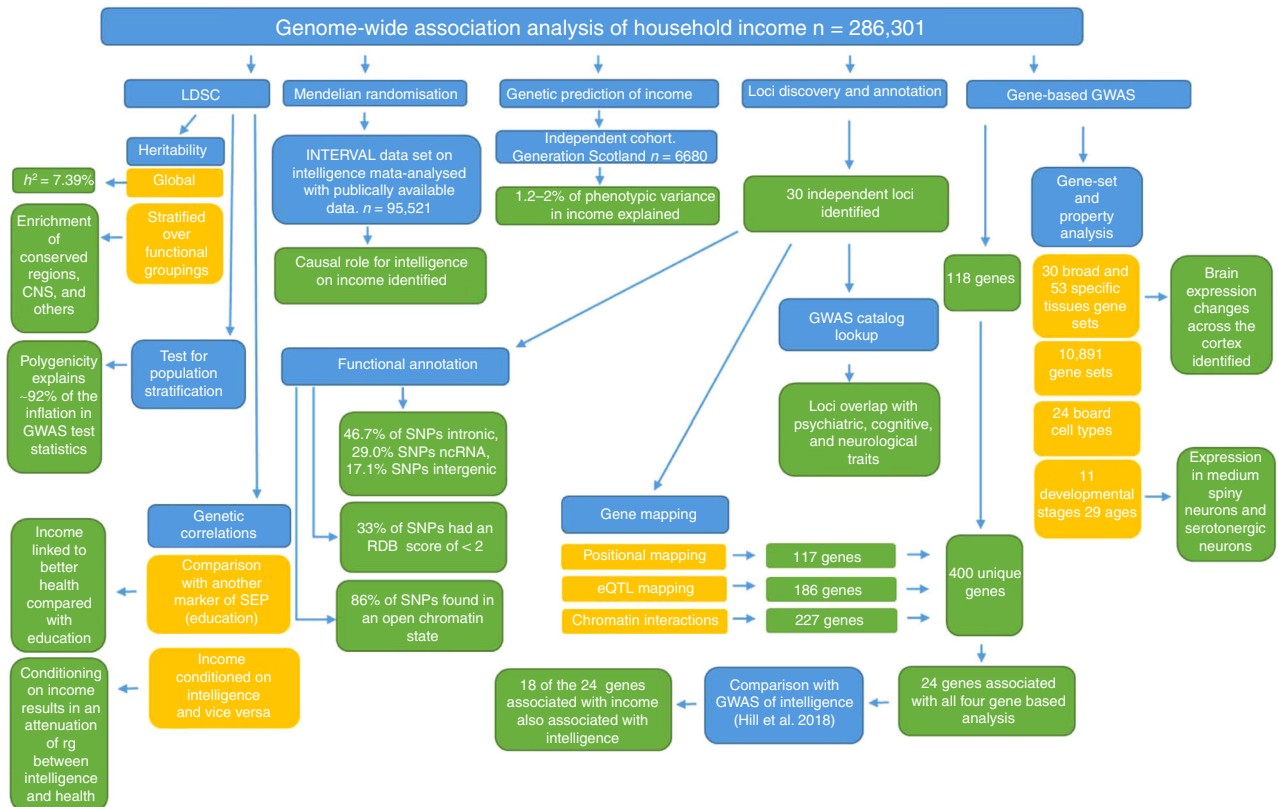

**Fig. 2 Flow chart for the statistical analysis carried out using the GWAS data on household income in 286,301 White British participants in UK Biobank.** Blue indicates a type of analysis conducted (i.e. LDSC to derive a heritability estimate) and gold indicates a subtype of this type of analysis (i.e. global heritability or the heritability of a subset of SNPs). Green indicates the result of an analysis (i.e. the global heritability was 7.39%).

total of 29 of these 30 loci were not identified in the previous UK Biobank analysis of income[7] (Supplementary Data 2). The 30 loci predominantly contained SNPs found within intronic regions (47%) as well as non-coding RNA introns (29%). A total of 17% of the SNPs within the independent loci were found in intergenic regions, and only 1.2% were found in exons (Fig. 3b). Many of the loci contained SNPs showing evidence of influencing gene regulation with 33% having a Regulome DB score of <2 (Fig. 3c) and 86% having evidence of being in an open-chromatin state (indicated by a minimum chromatin state of <8, in Fig. 3d). Additionally, these loci were linked to intelligence (11 loci), mental health (schizophrenia, 1 locus; bipolar disorder, 2 loci; neuroticism, 4 loci) and neurological variables (corticobasal degeneration, 1 locus; subcortical brain volumes, 1 locus; and Parkinson's disease, 1 locus) (Supplementary Data 3).

Linkage disequilibrium score (LDSC) regression showed that the mean $\chi^2$ statistic was 1.45 and the intercept of the LDSC regression was 1.04. These statistics indicate that around 92% of the inflation in the GWAS test statistics was due to a polygenic signal rather than residual stratification or confounding. The LDSC regression estimate of the heritability of household income was 7.39% (SE = 0.33%).

**Gene prioritization.** Three methods of mapping allelic variation to genes were used to better understand the functional consequences of the 30 independent loci linked to income (positional mapping, expression quantitative trait loci (eQTL) analysis and chromatin mapping). Using positional mapping, SNPs from the GWAS were aligned to 117 genes. eQTL mapping was used to match *cis*-eQTL SNPs to 186 genes, and chromatin interaction mapping linked the SNPs to 277 genes (Fig. 3e and Supplementary

Data 4 and Supplementary Fig. 1). These mapping strategies identified a total of 400 unique genes, of which 133 (Fig. 3e cells 14 + 23 + 26 + 3 + 24 + 11 + 2 + 30) were implicated by at least two mapping strategies and 47 (Fig. 3e cells 23 + 24) were implicated by all three. Of the 133 implicated by two mapping strategies, two showed evidence of a chromatin interactions with two independent genomic risk loci (Supplementary Data 5). Both *HOXB2* and *HOXB7* showed interactions with loci 24 and loci 25. *HOXB2* showed interactions in mesendoderm (an embryonic tissue layer) tissue and IMR90 (foetal lung fibroblasts) tissue, whereas *HOXB7* showed associations in the tissues of hESC (human embryonic stem cell), mesenchymal (multipotent stromal cells which differentiate into a variety of different cell types) stem cell, IMR90, left ventricle, GM12878, and trophoblast-like cells.

**Gene-based association analysis.** Using MAGMA[13], 118 genes were associated with income ($P < 2.662 \times 10^{-6}$) (Supplementary Data 6 and Fig. 4a). These genes overlapped with 24 of those implicated using positional, eQTL, and chromatin interaction modelling (Fig. 3e). Of the genes implicated by each of the three methods and the gene-based GWAS, *BSN* was of particular note due to it being expressed primarily in the neurons of the brain and its role in the scaffolding protein involved in the organization of the presynaptic cytoskeleton. Also found in this overlap was the gene *CHST10*. The protein encoded by *CHST10* is a sulfotransferase that acts on HNK-1, which is involved in neurodevelopment and synaptic plasticity.

These 24 genes were then examined to determine if gene-based statistics had implicated them in intelligence due to the previously reported strong genetic correlations between income and intelligence[7]. We found that 18 were associated ($P < 2.75 \times 10^{-6}$)

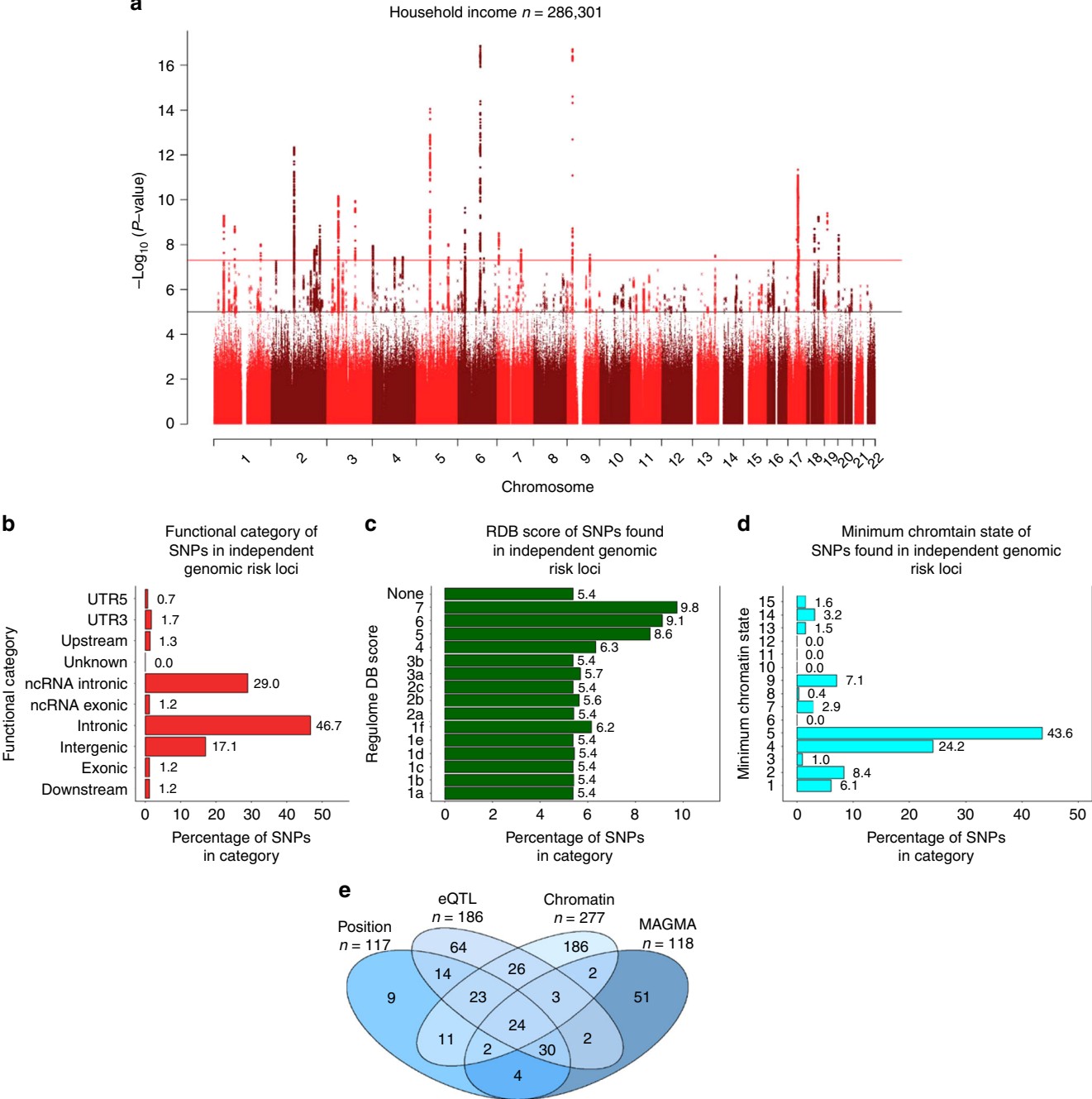

**Fig. 3 SNP level associations for income and mapping of the SNPs in independent genomic loci. a** Manhattan plot for income; negative log 10-transformed *P* values for each SNP are plotted against chromosomal location. The red line indicates genome-wide significance and the black line indicates suggestive associations ($1 \times 10^{-5}$). **b** Functional annotation carried out on the independent genomic loci identified. The percentage of SNPs found in each of the nine functional categories is listed. **c** The percentage of SNPs from the independent genomic loci that fell into each of the Regulome DB scores categories. A lower score indicates greater evidence for that SNPs involvement in gene regulation. **d** The percentage of SNPs within the independent genomic loci plotted against the minimum chromatic state for 127 tissue/cell types. **e** Venn diagram illustrating the overlap of the genes implicated using positional mapping, eQTL mapping, chromatin interaction mapping, which was conducted on the independent significant loci identified in the SNP-based GWAS. Also shown is how these implicated genes overlap with those identified using the gene-based statistics derived using MAGMA.

with intelligence from the GWAS conducted by Hill et al.[8]. This indicates that the genes with the most biological relevance to income were also linked to intelligence, again suggestive of the role that intelligence plays in SEP differences.

**Gene-set and gene-property analysis.** Gene-set analysis did not reveal evidence that any of the gene sets included here were enriched for differences in household income (Supplementary Data 7). However, a gene-property analysis showed that genes that were more associated with household income were also more highly expressed in the brain ($P = 1.31 \times 10^{-5}$) and the testis ($P = 1.31 \times 10^{-5}$) than genes that were less associated with income (Supplementary Table 1). This relationship was found across tissues of the cerebellum ($P = 5.61 \times 10^{-6}$), the cerebellar hemisphere ($P = 5.99 \times 10^{-6}$), the frontal cortex BA9 ($P = 9.68 \times 10^{-5}$), the cortex

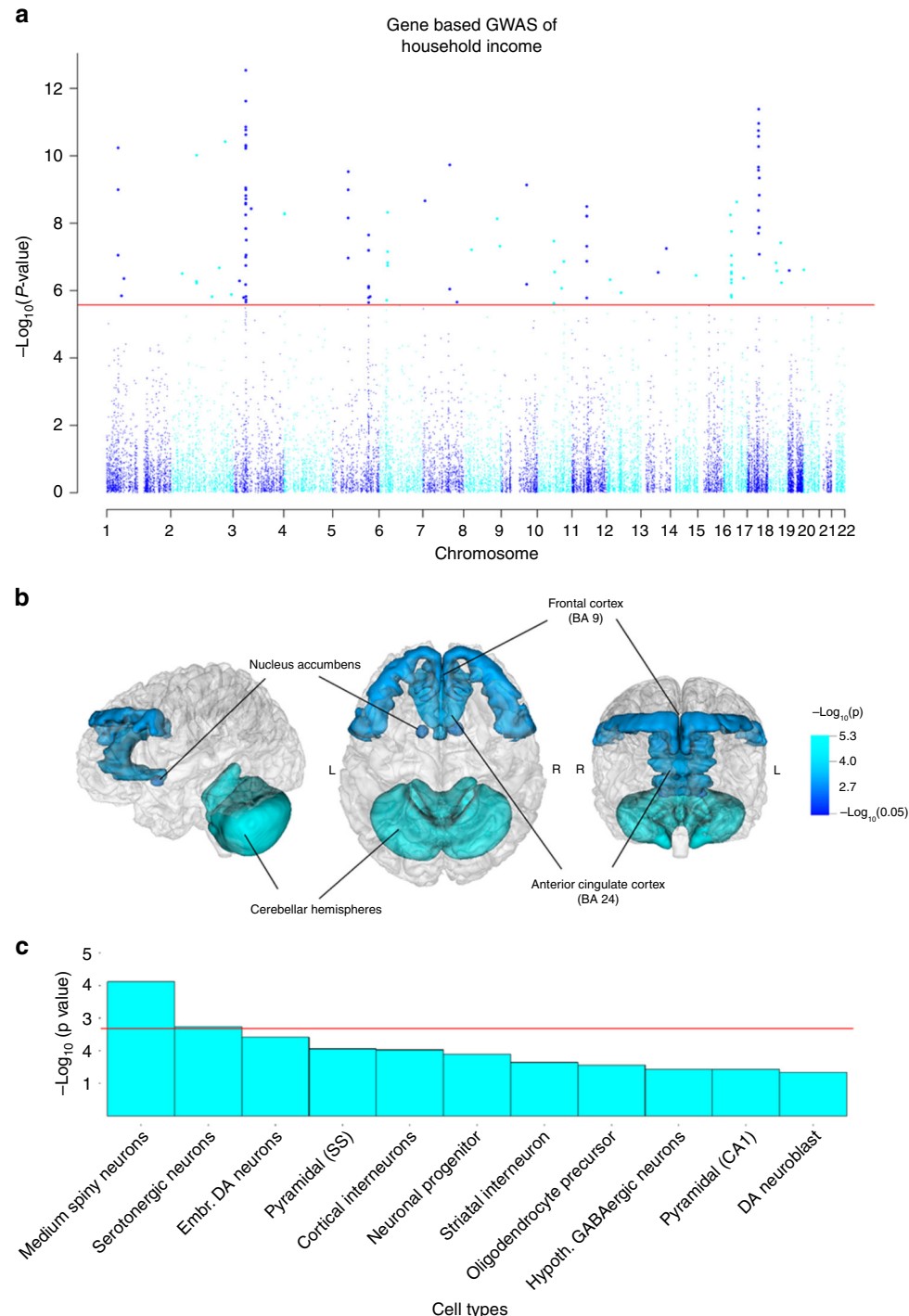

**Fig. 4 Gene-level associations for income and links to transcription in the brain and cortical cell types. a** A Manhattan plot of income using the gene-based statistics derived using MAGMA; $-\log_{10}$-transformed $P$ values for each gene are plotted against chromosomal location. The red line indicates genome-wide significance. **b** The results of a gene-property analysis linking transcription differences in the brain with income differences. Significant links between expression differences in cerebellar hemisphere, at Brodmann area 9 (BA9) of the frontal cortex, the nucleus accumbens and at Brodmann area 24 of the anterior cingulate cortex (BA24) are illustrated. Dark blue indicates low $-\log_{10} P$ values (a lower level of association) describing the link between gene expression and household income and light blue indicates high $-\log_{10} P$ values (a higher level of association) describing the same relationship. The full results are found in Supplementary Data 8 with the gene-based statistics produced using MAGMA. **c** shows the results of a cell-type-specific gene-property analysis where the relationship between the gene-based statistics from MAGMA and the degree to which gene expression was specific to the annotations was examined. A Bonferroni correction was applied to control for the 24 tests conducted. The red line indicates statistical significance indicating that expression that is specific to the annotation is associated with the gene-based statistics for income. Embr. DA Neurons, Embryonic Dopaminergic Neurons; Pyramidal (SS), Pyramidal (Somatosensory); Hypoth. GABAergic Neurons, Hypothalamic GABAergic Neurons; DA Neuroblast, Dopaminergic Neuroblast.

($P = 1.05 \times 10^{-4}$), the nucleus accumbens basal ganglia ($P = 2.93 \times 10^{-4}$) and the anterior cingulate cortex BA24 ($P = 6.81 \times 10^{-4}$) (Supplementary Data 8 and Fig. 4b).

Cell-type analysis conducted for household income indicated that of the 24 cell types examined, two were statistically significant after controlling for 24 tests. The significant cell types include medium spiny neurons $P = 7.67 \times 10^{-5}$, and serotonergic neurons $P = 0.002$ (Supplementary Table 2 and Fig. 4c). Finally, gene-property analysis found little evidence that genes linked to household income were transcribed in the brain at any one of 11 developmental stages[14], or across 29 different specific ages[14] (Supplementary Tables 3 and 4).

**Partitioned heritability**. The partitioned heritability analysis describes whether or not the SNPs that capture the greatest proportion of the heritability of income, also cluster in regions of the genome that are united by a shared biological theme. We find that, consistent with the notion that intelligence and income are genetically linked[15], the regions of the genome that have undergone purifying selection are those that harbour the greatest proportion of heritability for income ($P = 1.62 \times 10^{-10}$). Also enriched was the Conserved (GERP RS ≥ 4) annotation providing further evidence that conserved regions of the genome are enriched for the heritability of income. None of the other functional categories were significantly enriched for the heritability of income (Fig. 5a and Supplementary Data 9).

The partitioned heritability analysis using the six continuous categories analyzed by quintile showed that common variants that were in the first three quintiles for age (i.e. the younger three groupings) were associated with a greater proportion of the heritability of income (1st quintile $P = 2.57 \times 10^{-4}$, 2nd quintile $P = 3.33 \times 10^{-7}$ and 3rd quintile $P = 6.91 \times 10^{-16}$) as were SNPs in the upper two quintiles for greater level of background selection (4th quintile $P = 9.81 \times 10^{-8}$, 5th quintile $P = 0.001$). The first three quintiles describing nucleotide diversity and the same quartiles describing the level of LD (LDD – AFR) were also significantly enriched for heritability (nucleotide diversity, 1st quintile $P = 2.47 \times 10^{-23}$, 2nd quintile $P = 3.79 \times 10^{-20}$ and 3rd quintile $P = 0.003$; LDD–AFR, 1st quintile $P = 5.38 \times 10^{-12}$, 2nd quintile $P = 7.36 \times 10^{-16}$ and 3rd quintile $P = 0.002$) (Fig. 5b and Supplementary Table 5). The enrichment found by examining the continuous annotations by quintile is consistent with the idea that negative selective pressure has acted on the partially heritable traits linked to income.

When examining cell-type-specific enrichment using partitioned heritability, we show that the greatest level of enrichment for cell-type-specific groupings comes from the brain and central nervous system. This is indicated by the fact that the 24 cell types that were significantly enriched using the gene expression data set were all cell types that are found within the brain and the rest of the central nervous system (Fig. 5c and Supplementary Data 10). In addition, using the chromatin-based sets, 32 of the 34 cell groupings that were significantly enriched were drawn from the brain and the central nervous system (Fig. 5d and Supplementary Data 11).

This enrichment of heritability in the central nervous system led us to examine brain regions and cell types. We found that gene expression in the cortex harboured an enriched proportion of the heritability of income ($P = 0.006$), but no other regions were found to be enriched (Fig. 5e and Supplementary Table 6). Finally, gene expression associated with the category of neuron was found to be enriched ($P = 1.30 \times 10^{-9}$), but the two glia annotations of astrocyte and oligodendrocyte were not linked to income (Fig. 5f and Supplementary Table 7).

**Inference of causal links with intelligence**. Mendelian randomization (MR) was performed using the genetic instrument

derived using 19 SNPs associated with intelligence from a meta-analysis of a GWAS of intelligence from the INTERVAL study[16,17] as well as publicly available sources (Supplementary Methods). Here we inferred a strong, causal link between intelligence and income ($\beta = 0.213$, SE = 0.063, $P = 7.63 \times 10^{-4}$) (Supplementary Table 8). Should the assumptions of MR be met, this indicates that greater intelligence causes a higher level of income. Sensitivity analyses revealed little evidence of directional pleiotropy, which can bias MR estimates (MR–Egger intercept = 0.010, SE = 0.007, $P = 0.189$) (Supplementary Table 8). The heterogeneity statistics indicate that the estimated size of the causal effect of intelligence on income varies across the SNPs (Supplementary Table 8). However, since there was little evidence of directional pleiotropy, the overall causal estimate based on all of the genetic variants is unlikely to be biased if the MR–Egger assumptions hold (i.e. the InSIDE assumption).

**Genetic correlations**. Genetic correlations were calculated between household income and a set of 27 data sets covering psychological traits, mental health, health and well-being, anthropometric traits, metabolic traits and reproduction.

First, we build on the findings of Hill et al. (2016)[7] by using a larger, better-powered data set on income to show that the genetic variants associated with household income are linked with those that influence intelligence, $r_g = 0.69$, SE = 0.02, $P < 10 \times 10^{-200}$. We also show that there are genetic correlations between income with health (self-rated health, $r_g = 0.60$, SE = 0.03, $P = 5.72 \times 10^{-73}$), mental health (subjective well-being, $r_g = 0.32$, SE = 0.04, $P = 4.99 \times 10^{-17}$) and longevity ($r_g = 0.47$, SE = 0.07, $P = 1.29 \times 10^{-10}$). Furthermore, we replicated the finding of Hill et al. (2016)[7] by showing in our current study that while a general factor of neuroticism shows a negative genetic correlation with household income ($r_g = -0.36$, SE = 0.02, $P = 2.07 \times 10^{-53}$), the two special factors of neuroticism named anxiety/tension and worry/vulnerability each show positive genetic correlations with income ($r_g = 0.12$, SE = 0.03, $P = 7.19 \times 10^{-5}$ and $r_g = 0.15$, SE = 0.03, $P = 5.61 \times 10^{-7}$, respectively).

These findings show that many of the same genetic variants linked to higher SEP, as measured by income, are also linked to better health. It should, however, be noted that income shows a positive genetic correlation with the mental health variables of anorexia nervosa ($r_g = 0.09$, SE = 0.03, $P = 9.53 \times 10^{-3}$) and bipolar disorder ($r_g = 0.11$, SE = 0.04, $P = 1.20 \times 10^{-2}$) (Fig. 6a and Supplementary Data 12).

Second, as SEP is a multi-dimensional construct and each marker of SEP is imperfectly correlated with the others, the magnitude of the genetic correlations derived using income was compared with those derived using another measure of SEP, educational attainment. The goal of these analyses was to indicate if the genetic associations between income with health differed from those of education with health. As can be seen in Fig. 6a, whereas the magnitude and direction of the genetic correlations derived using income and educational attainment with the 27 health and well-being, anthropometric, mental health and metabolic traits were highly similar, there were instances of divergence indicating unique genetic associations with the two SEP variables. Of note are the variables of autism and schizophrenia. As found in previous studies[8,18–23], schizophrenia showed a small positive genetic correlation with educational attainment ($r_g = 0.06$, SE = 0.02, $P = 1.15 \times 10^{-3}$), whereas, in the present study, income showed a negative genetic correlation with schizophrenia ($r_g = -0.14$, SE = 0.02, $P = 6.49 \times 10^{-9}$, $P_{diff} = 6.57 \times 10^{-11}$). Autism was positively genetically correlated with educational attainment ($r_g = 0.27$, SE = 0.03, $P = 1.10 \times 10^{-15}$) as previously described[8,21,24], whereas there was no detectable genetic correlation between income and autism

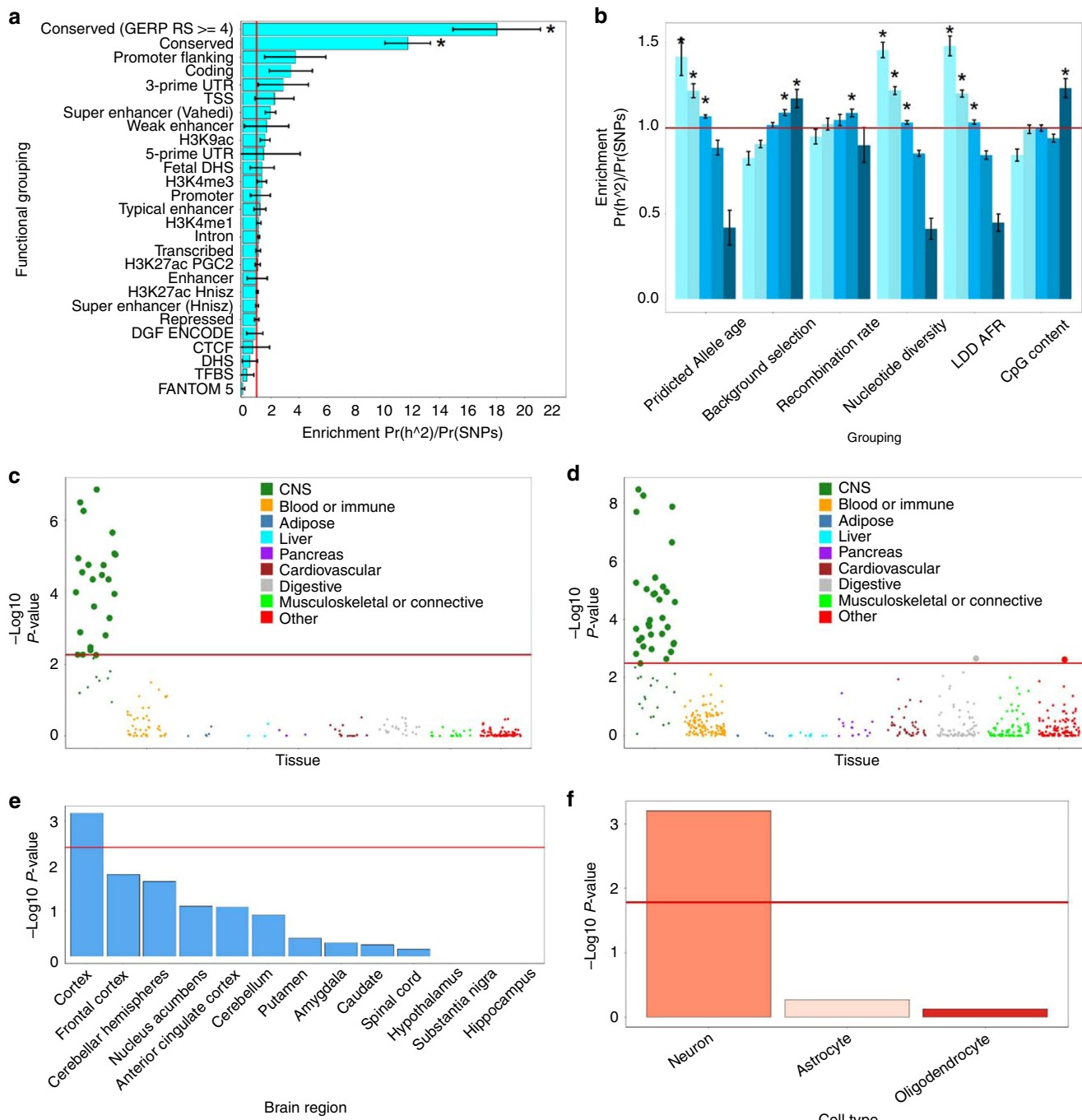

**Fig. 5 Partitioned heritability of income. a** Enrichment analysis for income using the 60 functional categories as well as 10 minor allele frequency groupings and 6 continuous annotations (27 categories describing enrichment within these categories is shown). This analysis differs from that presented in Figs. 3 and 4 as here all SNPs are used, not only those that reached genome-wide significance (Fig. 3) or SNPs that were located within protein-coding genes (Fig. 4). The enrichment statistic is the proportion of heritability found in each functional group divided by the proportion of SNPS in each group (Pr $(h^2)$/Pr(SNPs)). The red line indicates no enrichment found when Pr $(h^2)$/Pr(SNPs) = 1. Error bars represent ± 1 SE. A Bonferroni correction controlling for 57 tests was used to ascertain statistical significance that is indicated by an asterisk. **b** Enrichment analysis for the six continuous annotations by quintile. Shading represents quintile with light colours corresponding to low quintiles and dark colours to high quintiles. Groupings that contained a significantly greater proportion of heritability proportional to the number of SNPs they contain are marked with an asterisk. Multiple testing was performed within each of the annotations resulting in an $\alpha$ level of $\alpha = 0.05/5 = 0.01$, with a red line indicating no enrichment. Error bars represent ± 1 SE. **c**, **d** shows the enrichment of 205 tissues of cell types assembled using gene expression data and 489 groupings assembled using chromatin data. In each instance, these were arranged into nine tissue-type groupings with correction for multiple testing been performed using false discovery rate (FDR)[83] conducted separately for the gene expression and the chromatin groupings indicated by a red line. **e** Shows if the genes that were expressed in 13 brain regions are enriched for the heritability of income. A Bonferroni correction was used to control for 13 tests and the $\alpha$ level was set at 0.004 with the brain regions that crossed the red line being those that were statistically significant. **f** shows the level of enrichment for three brain cell types. A Bonferroni correction was used to control for three tests ($\alpha = 0.05/3 = 0.017$) and groupings that crossed the red line were those that were statistically significant. The full results for each of these analyses can be found in Supplementary Data 9, Supplementary Table 5, Supplementary Data 10 and 11 and Supplementary Tables 6 and 7.

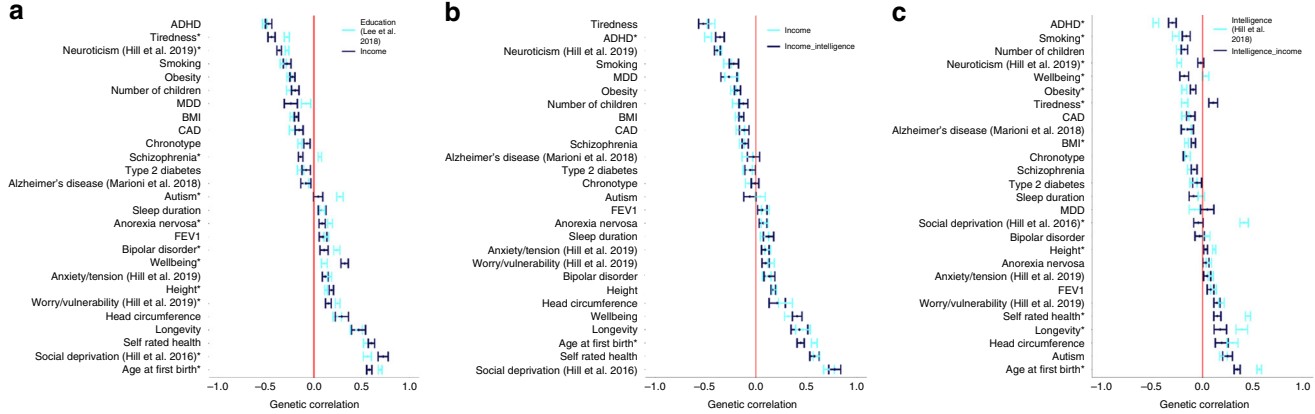

**Fig. 6 Pairs of genetic correlations are compared. a** compares genetic correlations derived using income with those derived using education, **b** compares genetic correlations derived using income with those derived using income conditioned on intelligence and **c** compares genetic correlations derived using intelligence with those derived using intelligence conditioned on income. In each instance, 27 pairs of genetic correlations are compared. Genetic correlations that were significantly different between within each of the three comparisons described above using a two-sided test ($2*\text{pnorm}(-\text{abs}(\text{abs}(r_{gi} - r_{gj})/\text{sqrt}(SE_i^2 + SE_j^2))))$) are indicated by an asterisk next to the phenotype label. MDD, major depressive disorder; ADHD, attention-deficit hyperactivity disorder; T2D, type 2 diabetes; CAD, coronary artery disease; SRH, self-rated health; SWB, subjective well-being; BMI, body mass index: FEV1, forced respiratory volume in 1 second. Worry/vulnerability and anxiety/tension were derived as special factors of neuroticism[34,92]. Full results for each of the genetic correlations derived can be found in Supplementary Table 12. Error bars represent ±1 SE.

($r_g = 0.04$, SE $= 0.05$, $P = 0.37$, $P_{\text{diff}} = 1.17 \times 10^{-11}$). There was evidence of differences between the income and education genetic correlations and nine other traits (subjective well-being, $P_{\text{diff}} = 1.42 \times 10^{-5}$, tiredness, $P_{\text{diff}} = 1.60 \times 10^{-4}$, age at first birth, $P = 1.24 \times 10^{-3}$, bipolar disorder, $P_{\text{diff}} = 1.41 \times 10^{-2}$, social deprivation, $P_{\text{diff}} = 1.72 \times 10^{-2}$ and chronotype, $P = 3.83 \times 10^{-2}$, the worry/vulnerability special factor of neuroticism $P = 1.17 \times 10^{-2}$ and a general factor of neuroticism $P_{\text{diff}} = 7.26 \times 10^{-3}$) (Fig. 6a and Supplementary Data 12).

Third, the role of intelligence in mediating the effect of genetic variation on income was explored by estimating the genetic correlation of income with each of the traits after conditioning the income GWAS on a GWAS of intelligence. As can be seen in Fig. 6b, after controlling for intelligence, the genetic correlations between income and the 27 health and well-being, anthropometric, mental health and metabolic traits remained largely similar. Two exceptions to this were age at first birth, where the genetic correlation with income decreased from $r_g = 0.58$ (SE $= 0.03$, $P = 8.81 \times 10^{-99}$) to $r_g = 0.45$ (SE $= 0.04$, $P = 1.20 \times 10^{-35}$, $P_{\text{diff}} = 0.003$), and ADHD, which decreased from $r_g = -0.48$ (SE $= 0.03$, $P = 2.20 \times 10^{-45}$) to $r_g = -0.36$ (SE $= 0.04$, $P = 1.86 \times 10^{-17}$, $P_{\text{diff}} = 0.03$). This means that genetic variation that is associated with income, but not intelligence, shows much of the same overlap with the 27 traits used here, as the genetic variation that is common to both income and intelligence.

In Fig. 6c however, 12 genetic correlations with intelligence changed after controlling for income. There was little evidence that subjective well-being was genetically correlated with intelligence ($r_g = 0.03$, SE $= 0.03$, $P = 0.31$), as previously found;[8] however, subjective well-being was negatively genetically correlated after adjusting for income ($r_g = -0.18$, SE $= 0.04$, $P = 3.11 \times 10^{-5}$) ($P_{\text{diff}} = 9.92 \times 10^{-5}$). The genetic correlation between intelligence and social deprivation (as measured by Townsend scores) of $r_g = -0.42$ (SE $= 0.04$, $P = 1.38 \times 10^{-23}$) attenuated to $r_g = 0.04$ (SE $= 0.05$, $P = 0.38$) ($P_{\text{diff}} = 1.29 \times 10^{-13}$). The genetic correlation between intelligence and neuroticism ($r_g = -0.23$, SE $= 0.02$, $P = 1.83 \times 10^{-23}$) also attenuated to close to zero after conditioning on income ($r_g = -0.02$, SE $= 0.03$, $P = 0.57$) ($P_{\text{diff}} = 7.26 \times 10^{-3}$). This means that the genetic variation that is associated with intelligence, but not income, shows less overlap with the 27 traits used here, than the genetic variation that is common to both

intelligence and income. The genetic correlations with intelligence once conditioning on income were different for the variables of self-rated health ($P_{\text{diff}} = 6.76 \times 10^{-12}$), age at first birth ($P_{\text{diff}} = 1.33 \times 10^{-8}$), fatigue or tiredness ($P_{\text{diff}} = 6.82 \times 10^{-8}$), ADHD ($P_{\text{diff}} = 5.55 \times 10^{-4}$), height ($P_{\text{diff}} = 2.59 \times 10^{-4}$), BMI ($P_{\text{diff}} = 0.013$), obesity ($P_{\text{diff}} = 1.60 \times 10^{-2}$), longevity ($P_{\text{diff}} = 0.014$) and smoking ($P_{\text{diff}} = 0.032$) (Fig. 6c, Supplementary Data 12).

**Genetic prediction.** Polygenic risk scores (PGRSs) were derived using the summary statistics from our GWAS of household income and GS:SFHS. When examining the PGRSs within each of the five income groups in GS:SFHS, we found that those in category 5 (those earning more than £70,000) had the highest PGRSs (Fig. 7a). The predicted income for the PGRSs was lower in each subsequent level of household income in GS:SFHS.

Those in the lowest quintile of the polygenic score for income were found on average to have the lowest predicted income (Fig. 7b), with the mean level of household income rising across each quintile. Those in the three lowest quintiles for their genetic propensity for income were found to have an average level of household income between £10,000 and £30,000, whereas those in the top two quintiles were found to have a household income of between £30,000 and £50,000. Polygenic prediction conducted using the summary data from UK Biobank applied to the GS: SFHS data showed that between 1.2 and 2.0% of the variance in household income can be predicted using the polygenic score for income (Supplementary Table 9 and Fig. 7c), with the PGRSs that was most predictive using a $P$ value cutoff of 0.1.

**Multi-trait analysis of GWASs.** MTAG has previously been used to conduct the first well-powered GWAS on intelligence[8]. We used MTAG here to increase the power of our GWAS on income by meta-analyzing it with another measure of SEP, educational attainment[10], as measured by the number of years of education a participant has completed. MTAG was conducted using the default settings and applied to increase the power in the GWAS of household income. Following the application of MTAG, the mean $\chi^2$ statistic increased from 1.45 to 1.73 and increased the effective sample size to 505,541 for income.

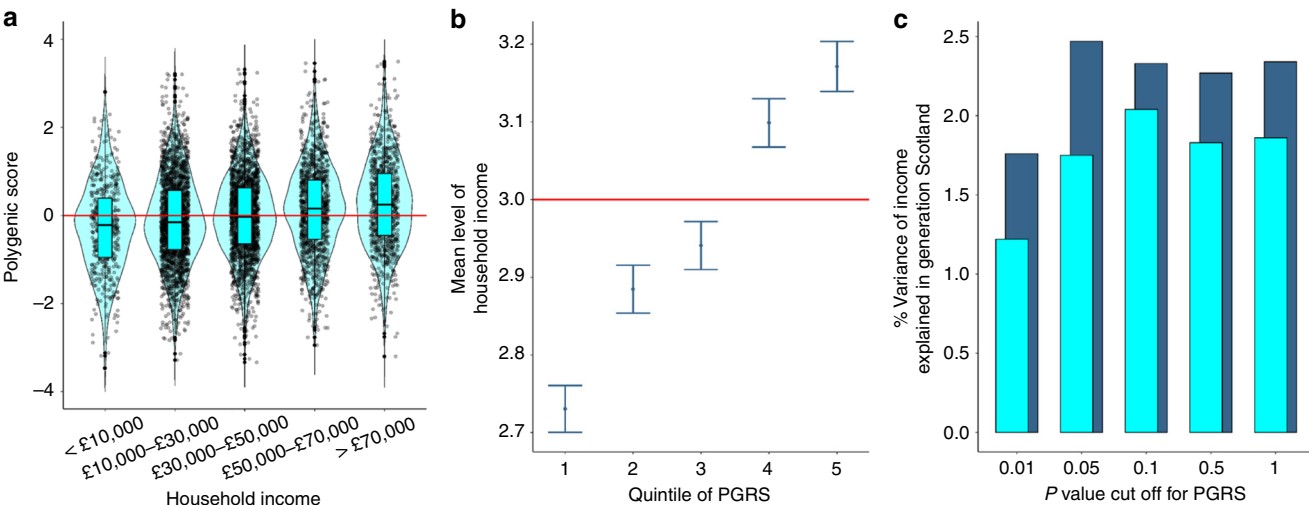

**Fig. 7 Polygenic risk score analysis of income. a** Violin plot showing the level of household income in GS:SFHS plotted against the standardized polygenic score of income in each group. Median and interquartile range are plotted. Summary data from the income GWAS performed in UK Biobank were used to derive PGRSs. Red line indicates a standardized polygenic score of 0. **b** The average level of household income for the five PGRSs is shown. Summary data from the income GWAS performed in UK Biobank were used to derive PGRSs. The y-axis corresponds to the 5-point classification of household income in Generation Scotland. Above the red line indicates a level of household income between £30,000 and £50,000 and below indicates a level of household income between £10,000 and £30,000 in Generation Scotland. Error bars represent ± 1 SE. **c** The variance accounted for by each of the five P value cutoffs for the PGRS. Light orange indicates that the income phenotype derived in UK Biobank was used to generate polygenic risk scores along with Generation Scotland. The dark orange bars indicate instances of where the MTAG phenotype derived using income and educational attainment was used to derive polygenic risk scores in Generation Scotland. Summary data from the income GWAS performed in UK Biobank, and the MTAG analysis of income was used to derive PGRSs. All results can be found in Supplementary Table 9.

The maxFDR derived was 0.003, over an order of magnitude lower than the commonly accepted standard of false discovery and comparable with that reported previously[8,25], indicating that the data set was capturing variance associated with income. We also found that the genetic correlation between our MTAG-income phenotype and a previous GWAS on income[7] was $r_g = 0.97$ (SE = 0.024), with a genetic correlation of $r_g = 0.94$ (SE = 0.004) with educational attainment. This indicates that the polygenic signal in the MTAG-income analysis is virtually identical to that found in previous GWAS of income, but also that it captures more of the variance that is shared between income and education.

Using this MTAG-income phenotype, we identify 144 independent genomic risk loci (Supplementary Fig. 2A and Supplementary Data 13). A total of 24 overlapped with the 30 found without using MTAG, meaning that by using MTAG, an additional 120 independent loci were identified that were associated with income (Supplementary Data 14). Functional annotation of these loci, as well as gene-based analyses and partitioned heritability analysis, showed results that were consistent with a better-powered GWAS data set on household income (Supplementary Fig. 2B–E). These results can be found in Supplementary Note 1.

PGRS analysis using the MTAG phenotype showed that between 1.7 and 2.5%, the variance of income was predicted in an independent sample (Supplementary Table 9 and Fig. 7c), with the PGRS that was most predictive using a P value cutoff of 0.05.

## Discussion

Using the UK Biobank data set, we identified 30 independent genetic loci (29 of which have not been previously reported) associated with income levels in Great Britain today. This represents a considerable advance on the two loci previously identified by Hill et al.[7] The present study contributes to our

understanding of the genetic contributions to SEP in at least seven major ways.

First, the loci associated with income showed clear evidence of functionality, particularly regarding their links to gene expression, regulatory regions of the genome and open-chromatin states. Second, by combining our GWAS data with eQTL data from BRAINEAC[26], GTEx[27] and others, along with chromatin inter-action data[28,29], we were able to prioritize which genes were likely to be causal based on the overlap of multiple lines of biological enquiry. Although income, as a biologically distal phenotype, will not be directly linked to genetic variation (Fig. 1)[7], genes that may exert a causal influence are likely to do so through their effect on more proximal phenotypes[30].

Using our GWAS data set on income, we identified 47 genes that were mapped to the 30 independent genomic loci using positional, eQTL and chromatin mapping. In addition, we used the 118 genome-wide significant genes from our gene-based analysis of income to further refine this set to a total of 24 implicated genes. These 24 genes therefore should be prioritized in follow-up studies as they are located close to the associated loci, have expression correlated with genetic variation of the SNPs in the independent genomic loci, and have chromatin interactions taking place between these genes and the SNPs found in the independent loci (Supplementary Data 4), and consistent with highly polygenic traits, these genes harbour many SNPs that show consistent associations with income (Supplementary Data 6). In addition, 18 of these genes have been associated with intelli-gence[8], so efforts to ascertain how such genetic variation is associated with income differences should examine their asso-ciations with intelligence more closely.

Third, by broadening our analysis to include the polygenic signal that fell outside the independent loci, we identified addi-tional, functional elements of the genome linked to differences in income. By combining the gene-based statistics from MAGMA with gene expression data from the GTEx[27] database, we

identified a positive association between expression in the brain and several specific regions, and the level of association displayed by the gene-based statistics on income. This indicates that the higher the level of association between a gene and income, the higher that gene's level of expression specific to the brain will be.

Cell-type-specific analysis revealed that the expression that was specific to the serotonergic neurons and to medium spiny neurons was associated with income. Medium spiny neurons have previously been linked to schizophrenia[31], which has a strong cognitive component and has previously been linked to gluta-matergic systems, including the $N$-methyl-D-aspartate receptor signalling complex[32]. Medium spiny neurons are a subtype of GABAergic inhibitory neurons. Future work should examine if, like other cognitive traits, income is linked to both GABAergic and glutamatergic systems.

Partitioned heritability analysis identified enrichment across cell types from the central nervous system and across the cortex as being significantly enriched for the heritability of income. These findings indicate that income in Great Britain today is associated with phenotypes that are associated with differences in the brain such as intelligence[8] or neuroticism[33,34].

These two approaches, gene-based statistics and LDSC regression, illustrate how combining the genetic data from GWAS with gene expression data can be informative as to the possible biological processes that are associated with income. This is of particular value for traits, like income, that have no clear biological analogue and are likely linked to genetic variation via mediated pleiotropy. This combination of data provides evidence that some of the individual differences in income are related to gene expression differences in the brain (Figs. 4b and 5c–e), as well as highlighting the role of specific classes of neurons (Figs. 4c and 5f). As importantly, we show the role for some tissue types outside of the central nervous system (Fig. 5d), indicating that genetic factors associated with income differences may also lie outside the phenotype of intelligence, and outside cortical tissue types.

Fourth, using MR, we provided evidence implicating intelligence as one of the potentially causal, partly heritable, phenotypes that might be one bridge in the gap between molecular genetic inheritance and phenotypic consequence. This result could help explain why individual differences in income are found to be partly heritable.

Fifth, our data show that income and education each have similar genetic correlations with many variables. However, some genetic correlations differ depending on whether income or education is used as a measure of SEP, and those that differed tend to be those related to mental health. In those, the income genetic correlations that are negative are of a greater magnitude than those derived using education, and where the income genetic correlations are positive, they are of smaller effect than the education-derived genetic correlations (Fig. 6a and Supplementary Data 12).

Together, this implies that the genetic variants that are associated with higher income tend to be more strongly associated with better psychological health than the genetic variants associated with education. This could be a stage-of-life-course-specific phenomenon, that is, education tends to be completed earlier in the life course, before some illnesses appear that could affect earning capability. It should also be considered that these significantly different genetic correlations between education and income indicate that educational attainment serves to provide access to opportunities in the labour market, and those that have these opportunities are then better placed to engage in health-relevant behaviours. This would indicate that whereas income may be a more distal phenotype from DNA than education, it is potentially closer to outcomes such as later-life health, as

evidenced by differences between the genetic correlations. Future work should examine models where DNA → neuronal properties → intelligence → education → income → health, using multi-variable MR[35–37] to gauge the direct and indirect effects of income and education on health outcomes.

However, previous work in Sweden using lotteries as natural experiments to examine the causal effect of wealth on health differences[38] found that in the 10 years after receiving a prize (either as a single payment or multiple instalments), winning participants did not have a longer life or fewer hospital admissions compared with those who did not win the lottery[38]. This indicates that whereas high earners may be in better health and have a greater level of education than low earners, a high income might not be causal in such differences in affluent countries that have strong social support systems. Furthermore, children born to lottery winners were not found to be advantaged in terms of their level of scholastic performance compared with the children of those who did not win the lottery[38], a finding that argues against a dynastic effect mediated via wealth. Although any causal effect of wealth on health is likely to differ across countries and time periods, should the results of this Swedish study generalize to the United Kingdom today, they would complement our results and together would support a model whereby genetic differences that are linked with health might be linked to partly heritable intermediary phenotypes, such as intelligence.

In this scenario, the similarities and differences between the genetic correlations derived using education and income might be accounted for in part by the differences in the intermediary phenotypes that give rise to each measure of SEP. Under this model, the observed differences between genetic correlations with mental health (Fig. 6a) would be due to intermediary variables that make a greater contribution to both income and mental health than they do to education. The similarities between income and education genetic correlations and health potentially indicate a similar contribution from intermediary phenotypes to income, education and health.

We found that when the genetic associations that are shared between income and intelligence were removed, the genetic correlations with other traits were largely unchanged. The exceptions were with ADHD and with age of first birth, where the genetic correlations with income are both attenuated once conditioned on intelligence. However, 12 of the income–health genetic correlations were attenuated after adjusting the SNP–income associations for intelligence. These results indicate that the genetic variation associated with intelligence and income is also associated with many health and mental health traits, because, when this shared variance is removed, leaving only the variance that is unique to intelligence, the magnitude of the genetic link between intelligence and health is reduced. In the case of the genetic link between intelligence, social deprivation, neuroticism and height, this genetic association disappears entirely following adjustment for income. The exception is that subjective well-being shows a genetic correlation with intelligence only after the variance that is common to both income and intelligence is removed.

One interpretation of this finding is that the residual variance left in income after conditioning on intelligence still contains the genetic contributions to other partly heritable traits (such as conscientiousness or resistance to disease). These traits also contribute towards individual differences in income, and so the association between income and health is, largely, intact following conditioning on intelligence. This would imply that intelligence is only one of a number of factors that contribute to variation in income, but income is a very important factor that mediates the associations between intelligence and health. Future work examining the genetic relationship between income and health, as well as intelligence and health, should focus on this genetic

overlap between intelligence and income using tools such as genomic structural equation modelling (SEM) to partition the total variance of traits like income into the variance that is shared with intelligence and the variance that is separate from it[39].

Sixth, we were able to predict up to 2% of income differences using PGRSs. This shows that even for phenotypes that are not impacted directly by genetic effects, but rather are more biologically distal as is the case with income, that the link between genotype and phenotype is sufficient to make predictions, based on DNA alone.

Seventh, using MTAG, we increased our effective sample size from 286,301 participants to 505,541. With this increase in power, we were able to increase the number of loci found to be associated with income from 30 to 144. Of these 144 associations, 120 were not found to be genome-wide significant before the application of MTAG. These loci demonstrated the same patterns of functional enrichment as shown in the 30 loci identified using income alone. We also identified the same relationship between expression in the brain and across multiple cortical structures, using the better-powered MTAG-derived income phenotype (Supplementary Note 1). Furthermore, following meta-analysis with MTAG, we were able to increase our prediction accuracy of income by 25%.

The limitations of this study include that income was measured at the level of the household and was not an individual-level measure of income. However, previous GWASs examining household income variables have shown that income, measured at a household level, has a genetic correlation of 0.90 (SE = 0.04) with educational attainment, as measured on an individual level, indicating that the household-level effects are likely to be generalizable to individual persons[7]. Furthermore, GWASs conducted on regional measures of educational attainment show genetic correlations of >0.9 with education measured using an individual's own level of educational attainment[40].

A limitation of the MR analysis specifically are potential dynastic effects, which may violate the assumptions of MR. Dynastic effects are where genetic variants that the parent has but the child does not, are associated with parental behaviours, and these parenting behaviours are a causal factor in the SEP of the child[41]. An example of this would be that parents with a greater predisposition towards intelligence are also those that are more likely to provide opportunities for their children to enter higher-income occupations. In this instance, the second assumption of MR, that the instrument must only affect income via their effect on intelligence, would be violated. The association of the offspring's SNPs and income would be partially due to the effects of the parents' genotype on their parents' intelligence, which subsequently affects offspring's income. Whereas the current data cannot differentiate between causality and dynastic effects, it should be noted that for another measure of SEP, educational attainment, there is evidence of indirect genetic effects that account for ~30% of the variance of the direct genetic associations[42]. Future work in multigenerational samples should examine the role that such indirect genetic effects play in individual differences in income, as well as if their presence (if established) could result in an inflation of the estimate for a causal effect using MR[43].

Furthermore, genetic variants associated with intelligence are likely to have pleiotropic effects[44]. However, to break the assumptions of MR, it is not sufficient for the genetic variants to have pleiotropic effects[45]. The genetic variants we use as instruments must have horizontally pleiotropic effects mediated via mechanisms other than intelligence. If the genetic variants have vertically pleiotropic effects, for example, SNP → neuron → intelligence → income → health, then our MR estimates will not be biased. Equally, if the SNPs affect other phenotypes, but these phenotypes do not affect outcome, then these effects will not

result in bias in the MR estimates. It is possible that the genetic variants identified in intelligence GWAS have horizontally pleiotropic effects; however, it is unclear what mechanisms would mediate these effects. The genetic correlations between intelligence and personality traits are relatively low[46]. The genetic variants identified in the intelligence GWAS are likely to also affect a range of cognitive ability-related traits. However again, these pleiotropic effects via related phenotypes are unlikely to cause bias if the results are interpreted as a test of general cognitive function. It is possible to investigate potentially horizontal pleiotropic effects further using multivariable MR[47]. If SNPs have been identified, which explain sufficient independent variation in two or more potential pathways, for example, intelligence and education, it is possible to identify the direct effects of each exposure. Future research should use multivariable MR to investigate this further.

Another limitation is that the present study was restricted to examining common genetic effects. Should rare or less common genetic variation be associated with income, then these effects will be absent from this study. Future work should utilize methods that can capture these genetic effects[48], as well as examine SEP variables using whole-exome or whole-genome sequencing. In addition, the participants of UK Biobank are drawn from the more educated and healthier individuals in the United Kingdom, which might introduce collider bias[49]. Whereas a comparison of the level of SEP between the individuals in UK Biobank and the census conducted in the United Kingdom indicates that SEP, as measured using the Townsend Deprivation Index[50], was very similar[7], future work aiming to quantify or control for collider bias would be of value in addressing this potential issue.

A further limitation is that molecular genetic analyses of phenotypes, such as intelligence, income or SEP, appear prone to being misinterpreted[51]. Such misunderstandings include describing associated variants as, genes for income, or the misinterpretation that any associated variant, and indeed any non-zero heritability estimate, is evidence for genetic determinism or the immutable nature of these phenotypes via environmental intervention. We include a figure (Fig. 1) that illustrates that genetic variants do not act directly on income; instead, genetic variants are associated with partly heritable traits (such as intelligence, conscientiousness, health, etc.), which have their own complex gene-to-phenotype paths (including neural variables) and are ultimately associated with income. Therefore, the genetic variant–income associations discovered here are no more for income than they are for these other traits. For more discussion of the implications of these results, aimed at the general reader, we have provided a Frequently Asked Questions (FAQ) document in Supplementary Note 2.

Finally, it should be noted that GWASs, like heritability estimates, describe differences that exist within populations. This means that although we report here that those with a greater number of intelligence-associated genetic variants tend to be those who report higher incomes, it does not hold that this is true across other societies or times. Indeed, the links between markers of SEP and health are not consistent across all societies[52]. Research into genetic links to education has found indications that the genetic variants linked to higher educational attainment are less predictive of success in societies that have less meritocratic selection for education and occupation[53]. Future work examining the relative contribution of genetic and environmental associations with income, as well as the biological systems causally implicated in any GWAS conducted on a marker of SEP across many cultures, would be valuable in identifying more and less meritocratic societies.

In conclusion, this work adds to the growing body of evidence indicating that markers of SEP, and their links to health, are likely

to be both genetic and environmental in origin[6,7]. We found that SEP variation in the Great Britain is partially accounted for by genetic differences in the population[54]. We found little evidence that these genetic differences were attributable to population stratification, but rather that they indicated the unequal distribution of heritable traits, including intelligence, across different SEP groups. Using multiple forms of biological data, we showed that these genetic differences are predominantly found in regions of the genome that have undergone negative selection and are related to differences in gene expression in the brain, particularly in medium spiny neurons. We also prioritize 24 genes for further follow-up and evidence from eQTL analysis, chromatin interactions, with previous associations of intelligence converging to implicate 18 of these genes. Furthermore, we identify intelligence as one of the likely causal psychological traits partly driving differences in income and SEP in Great Britain today.

## Methods

**Participants**. The primary sample used involved participants from UK Biobank, an open-access resource established to examine the determinants of disease in middle-aged and older adults living in the United Kingdom[55]. Recruitment to UK Biobank occurred between 2006 and 2010, targeting community-dwelling individuals from both urban and rural environments across a broad range of socioeconomic circumstances. A total of 502,655 participants were assessed at baseline on a range of cognitive and other psychological measures, physical and mental health, and their SEP. They donated a number of biological samples, including DNA for genotyping. In order to reduce the effects of population stratification, only participants from a single-ancestry group, those of White British ancestry, were included in the analysis. High-quality genotyping was performed by UK Biobank on 332,050 participants. Ethical approval for UK Biobank was received from the Research Ethics Committee (REC reference 11/NW/0382). This work was conducted under UK Biobank application 10279.

**Phenotype description**. A total of 332,050 participants had genotype data and data on their level of household income. Self-reported household income was collected using a 5-point scale corresponding to the total household income before tax, 1 being less than £18,000, 2 being £18,000–£29,999, 3 being £30,000–£51,999, 4 being £52,000–£100,000 and 5 being greater than £100,000. This 5-point scale was analyzed by treating the categories of income as a continuous variable. Participants were removed from the analysis if they answered "do not know" ($n = 12,721$), or "prefer not to answer" ($n = 31,947$). This left a total number of 286,301 participants (138,425 male) aged 39–73 years (mean = 56.5, SD = 8.0 years) with genotype data who had reported, between 1 and 5, their level of household income.

**UK Biobank genotyping**. Full details of the UK Biobank genotyping procedure have been made available[56]. In brief, two custom genotyping arrays were used to genotype 49,950 participants (UK BiLEVE Axiom Array) and 438,427 participants (UK Biobank Axiom Array)[56,57]. Genotype data on 805,426 markers were available for 488,377 of the individuals in UK Biobank. Imputation to the Haplotype Reference Consortium (HRC) reference panel leads to 39,131,578 autosomal SNPs being available for 487,442 participants[56]. Allele frequency checks[58] against the HRC[59] and 1000G[60] site lists were performed, and variants with minor allele frequencies (MAFs) differing more than ±0.2 from the reference sets were removed.

Additional quality control steps were conducted and described previously[8,34]. These included the removal of those with non-British ancestry based on self-report and a principal components analysis, as well as those with extreme scores based on heterozygosity and missingness. Individuals with neither XX nor XY chromosomes, along with those individuals whose reported sex was inconsistent with genetically inferred sex, were also removed. Finally, individuals with more than ten putative third-degree relatives (identified by Bycroft et al.[56] by estimating the kinship coefficients for all pairs of samples using the software KING[61]) were also removed. Following these exclusions, a sample of 408,095 individuals remained. Using GCTA–GREML on 131,790 reportedly related participants[62], related individuals were removed based on a genetic relationship threshold of 0.025. Following this quality control, household income data and genetic data were available on 286,301 participants. Following association analysis, SNPs with an MAF < 0.0005, and an imputation quality score <0.1 were removed. Finally, only biallelic SNPs were retained, resulting in 18,485,882 autosomal SNPs.

**GWAS in the UK Biobank sample**. The level of household income as measured on the 5-point scale was subjected to a regression using income as the outcome, as has been conducted previously[7], and 40 genetic principal components (to control for population stratification), genotyping array, batch, age and sex as predictors. The

residuals from this model were then used in a GWAS assuming an additive genetic model as implemented in BGENIE[56].

**Functional annotation and loci discovery**. Genomic risk loci were derived using the summary data from the data set of household income derived in UK Biobank, using FUnctional Mapping and Annotation of genetic associations (FUMA)[63]. First, independent significant SNPs were defined using a $P$ value cutoff of genome-wide significance ($P < 5 \times 10^{-8}$), as well as being independent from each other ($r^2 < 0.6$) within a 1-mb window. Second, SNPs that were in LD with any independent SNP ($r^2 \geq 0.6$) and within a 1-mb window in addition to being in the HRC genomes reference panel with an MAF >0.001, were included for further annotation. Third, lead SNPs were identified using the independent significant SNPs as defined above. Lead SNPs were a subset of the independent significant SNPs that were in LD with each other at $r^2 < 0.1$, with a 1-mb window. Fourth, genomic risk loci were created by merging lead SNPs if they were closer than 250 kb apart. This means that a genomic risk locus could contain multiple independent significant SNPs and multiple lead SNPs. Finally, all SNPs in LD of $r^2 \geq 0.6$ with one of the independent significant SNPs formed the border, or edge, of the genomic risk loci.

The lead SNPs and those in LD with the lead SNPs were then mapped to genes based on their functional consequences, as described using ANNOVAR[64] and the Ensemble genes build 85. Intergenic SNPs were annotated as the two closest flanking genes, which can result in them being assigned to multiple genes.

**Gene mapping**. Three strategies were used to link the income-associated independent genomic loci to genes. First, positional mapping[65] was used to map SNPs to genes based on physical distance. SNPs were mapped to genes if they were within 10 kb of a known protein gene found in the human reference assembly (hg19).

Second, eQTL mapping was carried out by mapping SNPs to genes if allelic variation at the SNP is associated with expression levels of a gene. For eQTL mapping, information on 45 tissue types from three databases (GTEx v7, Blood eQTL browser and BIOS QTL browser) based on cis-QTLs was used and SNPs were mapped to genes up to 1 Mb away. A false discovery rate (FDR) of 0.05 was used as a cutoff to define significant eQTL associations.

Finally, chromatin interaction mapping was carried out to map SNPs to genes when there is a three-dimensional DNA–DNA interaction between the SNP and gene. No distance boundary was applied as chromatin interactions can be long-ranging and span multiple genes. Hi-C data of 14 tissue types were used for chromatin interaction mapping[66]. In order to reduce the total number of genes mapped using chromatin interactions and to increase the likelihood that those mapped are biologically relevant, an additional filter was added. We only retained interaction-mapped genes if one region involved with the interaction overlapped with a predicted enhancer region in any of the 111 tissue/cell types found in the Roadmap Epigenomics Project[67], and the other region was located in a gene promoter region (i.e. 250 bp upstream and 500 bp downstream of the transcription start site and also predicted to be a promoter region by the Roadmap Epigenomics Project[67]). An FDR of $1 \times 10^{-5}$ was used to define a significant interaction.

**Gene-based GWAS**. Gene-based analyses have been shown to increase the power to detect association due to the multiple testing burden being reduced, in addition to the effects of multiple SNPs being combined[68]. Gene-based GWAS was conducted using MAGMA[13]. All SNPs located within protein-coding genes were used to derive a $P$ value describing the association found with household income. The NCBI build 37 was used to determine the location and boundaries of 18,782 autosomal genes, and linkage disequilibrium within and between genes was gauged using the HRC panel. In order to control for multiple testing, a Bonferroni correction was applied using each gene as an independent statistical unit ($0.05/18,782 = 2.66 \times 10^{-6}$). The gene-based statistics derived using MAGMA were then used to conduct the gene-set analysis, the gene-property analyses and the cell-type enrichment analysis.

**Gene-set analysis**. In order to understand the biological systems vulnerable to perturbation by common genetic variation, a competitive gene-set analysis was performed. Competitive testing, conducted in MAGMA[13], examines if genes within the gene set are more strongly associated with the trait of interest than other genes, and differs from self-contained testing by controlling for type 1 error rate as well as being able to examine the biological relevance of the gene set under investigation[69].

A total of 10,891 gene sets (sourced from Gene Ontology[70], Reactome[71] and MSigDB[72]) were examined for enrichment of household income. A Bonferroni correction was applied to control for the multiple tests performed on the 10,891 gene sets available for analysis.

**Gene-property analysis**. In order to identify the relative importance of particular tissue types, which may indicate the intermediary biological phenotypes that might act between genetic variation and SEP outcomes, a gene-property analysis was conducted using MAGMA. The goal of this analysis was to determine if, in 30 broad tissue types and 53 specific tissues, tissue-specific differential expression levels were predictive of the association of a gene with household income. Tissue types were taken from the GTEx v7 RNA-Seq (RNA-sequencing) database[73] with

expression values being log$_2$ transformed with a pseudocount of 1 after Winsorising at 50 with the average expression value being taken from each tissue. Multiple testing was accounted for using Bonferroni correction.

An additional gene-property analysis was conducted to determine if transcription in the brain at 11 developmental stages[14], or across 29 different age groupings[14], was associated with a gene's link to household income. This RNA-Seq GEncode v10 summarized to gene data was accessed using the following link: http://www.brainspan.org/api/v2/well_known_file_download/267666525. The detailed descriptions of the normalization processes used can be found in the technical white paper at http://help.brain-map.org/download/attachments/3506181/Transcriptome_Profiling.pdf?version=1&modificationDate=13820365 62736&api=v2, where a total of 524 samples were available for analysis. The developmental stages were assigned to each of the two groups (11 developmental stages and 29 age groupings) based on the age of the sample. The groupings of 25 post-conception weeks (pcw) and 35 pcw were excluded from the age groups as they contained fewer than three samples. Next, the 52,376 annotated genes were filtered so that the average reads per kilobase (RPKM) are >1. This was performed in the developmental group and in the age group separately. This resulted in 19,601 genes for the developmental- stage groupings and 21,001 genes for the age groupings. RPKM was then winsorized at 50 (RPKM >50 was replaced with 50). Then, the average of log-transformed RPKM with a pseudocount 1 (log$_2$(RPKM + 1)) per group (for either 11 developmental stages or 29 age groups) was used as a covariate conditioning on the average across all the labels. To control for multiple tests, a Bonferroni correction was used to control for 11 and 29 tests separately.

**Cell-type enrichment.** As previous studies had indicated the importance of cortical tissues to differences in SEP[7,10], a gene-property analysis was also conducted to examine a broad array of brain-specific cell types. Enrichment of heritability was tested against 173 types of brain cells (24 broad categories of cell types), which were calculated following the method described in Skene et al.[31]. Briefly, brain cell-type expression data were drawn from single-cell RNA-Seq data from mouse brains. For each gene, a specificity value for each cell type was calculated by dividing the mean Unique Molecular Identifier (UMI) counts for the given cell type by the summed mean UMI counts across all cell types. MAGMA[13] was used to calculate cell-type enrichments where specificity values were then divided into 40 equal-sized bins for each cell type for the MAGMA analysis. A linear model was fitted over the 40 specificity bins (with the least specific bin indexed as 1 and the most specific as 40). This was done by passing the bin values for each gene using the '--gene-covar onesided' argument.

**Univariate LDSC.** Univariate LDSC regression was performed on the summary statistics from the GWAS on household income in order to quantify the degree to which population stratification may have influenced these results.

For the GWAS on household income, LD score regression was carried out by regressing the GWA test statistics ($\chi^2$) from each GWAS onto the LD score (the sum of squared correlations between the MAF count of a SNP with the MAF count of every other SNP) of each SNP. This regression allows for the estimation of heritability from the slope, and a means to detect residual confounders using the intercept.

LD scores and weights were downloaded from http://www.broadinstitute.org/~bulik/eur_ldscores/ for use with European populations. SNPs were included if they had an MAF of >0.01 and an imputation quality score of > 0.9. Following this, SNPs were retained if they were found in HapMap 3 with MAF > 0.05 in the 1000 Genomes EUR reference sample. Following this, indels and structural variants were removed along with strand-ambiguous variants. SNPs whose alleles did not match those in the 1000 Genomes were also removed. The presence of outliers can increase the standard error in LDSC regression, and so SNPs where $\chi^2 > 80$ were also removed.

**Partitioned heritability.** Partitioned heritability was performed using stratified LDSC regression[74,75]. Stratified LD scores were calculated from the European-ancestry samples in the 1000 Genomes project and only included the HapMap 3 SNPs with an MAF of >0.05. The model was constructed using 60 overlapping, functional categories. In addition, ten MAF bins and six continuous annotations were included to control for LD-related bias in the partitioned heritability analysis by modelling regional LD, as well as MAF. Correction for multiple testing was performed using a Bonferroni test on the 60 functional categories ($\alpha = 0.00083$). The continuous annotations were also analyzed by examining the enrichment of each quintile for the six continuous categories of predicted allele age, background selection, recombination rate, nucleotide diversity, low levels of linkage disequilibrium in African populations and CpG content. Here, control for multiple testing was performed using a Bonferroni correction within each of the six annotations ($\alpha = 0.05/5 = 0.01$).

Cell-type analysis was conducted using the method of Finucane et al.[76]. Here, four data sets were used and examined for enrichment of household income. The first data set (gene expression) contained gene expression data from across 205 tissue or cell types taken from the GTEx[73] database and from Franke lab data set[77,78] from Finucane et al.[76]. The second data set (chromatin) contained data on

489 tissue and cell types taken from Roadmap Epigenomics consortium[67] and from EN-TEx, a subgroup of ENCODE[76,79]. Data pertaining to expression in 13 regions, the brain was taken from GTEx[73] and gene expression specific to the neuron, the astrocytes and the oligodendrocytes were taken from mouse data from the work of Cahoy et al.[80].

Multiple testing for the partitioning of the heritability by cell types was conducted using a Bonferroni correction across the 13 brain regions ($\alpha = 0.05/13 = 0.004$) and across the three types of neurons ($\alpha = 0.05/3 = 0.017$). For the gene expression and chromatin groupings, an FDR[81] was applied to the 205 tests performed to look at enrichment using gene expression ($\alpha = 0.006$) and to the 489 tests examining chromatin-based annotations ($\alpha = 0.003$).

**Mendelian randomization.** The causal effects of intelligence (termed the exposure in an MR analysis) on income (termed the outcome in an MR analysis) were investigated using univariate MR analysis. Here, the total causal effect of intelligence on income was examined by combining summary GWAS test statistics for intelligence and for income using an inverse-variance-weighted (IVW) regression model[82]. This is equivalent to a weighted regression of the SNP-outcome coefficients on the SNP-exposure coefficients, with the intercept constrained to zero (i.e. assuming no unbalanced horizontal pleiotropy).

The results of the IVW regression model were compared with the results obtained using MR–Egger regression[83]. MR analyses, which use multiple SNPs, are more likely to include invalid SNPs with horizontally pleiotropic effects[84]. By not constraining the intercept to zero (as done using IVW regression) MR–Egger relaxes the assumption that the effects of genetic variants on the outcome act solely through the exposure (in this case intelligence). The intercept parameter of the MR–Egger regression indicates the average directional pleiotropic effects of the SNPs on the outcome. As such, the direct pleiotropic effect that the SNPs have on the outcome, independent of the exposure, can be quantified, where a nonzero intercept provides evidence for bias due to directional pleiotropy and a violation of the MR IVW estimator assumptions. Of note is that the MR–Egger regression estimates only remain consistent if the magnitude of the gene-exposure associations, across all variants, are independent of their horizontally pleiotropic effects on the outcome (i.e. the InSIDE assumption holds)[83]. In addition, power is almost always lower for MR–Egger and it requires variation in the size of effect of the SNPs on the exposure (i.e. if all SNPs have similar sized effects on the exposure, then MR–Egger will have very low power).

For use with MR, two independent groups ($n = 95,521$ for intelligence and $n = 271,732$ for income) were created, whereby the GWAS on income was rerun using only those participants whose data were not included in the interim release of the UK Biobank genotype data. A GWAS data set on intelligence was created by meta-analyzing publicly available data on intelligence with a GWAS (conducted for this study) using data from the INTERVAL study[16,17] (Supplementary Methods) where 19 SNPs were identified as being genome-wide significant and independent. These 19 SNPs were used as instrumental variables for intelligence in the MR analysis.

**Genetic correlations.** Genetic correlations were derived using bivariate LDSC regression. A total of 27 GWAS data sets on health, anthropometric, psychiatric and metabolic traits were examined for a genetic correlation with income (Supplementary Table 16). Genetic correlations were also derived between household income with education and intelligence. There were three objectives to our analysis examining genetic correlations using household income. First, we sought to replicate the results of Hill et al.[7], who found genetic correlations between household income and other variables in a smaller data subset from the UK Biobank sample used here. Second, SEP is multi-dimensional in nature: it is composed of multiple measures, each of which are correlated imperfectly with the others. Because of this, different measures of SEP may have genetic variance that is both unique to them, and differentiates them from the others in the way it associates with health. To examine this, we compare how genetic correlations with household income and 27 health, anthropometric, psychiatric, cognitive and metabolic traits differed compared with the genetic correlations derived using a different, individual-level measure of SEP, that is educational attainment as measured by the number of years one has spent in education[11]. Third, Hill et al.[7] also found that the phenotypes with the strongest genetic correlations with income are those that are cognitive (verbal numerical reasoning, childhood IQ and years of education) in nature[7]. The magnitude of these genetic correlations might indicate the phenotypes that occur as potential mediators between molecular genetic inheritance and household income.

In addition, intelligence is known to be genetically correlated with many physical and mental health traits[18,21,85]. The role that intelligence might play in accounting for some of the genetic links between household income and 27 health and well-being, anthropometric, mental health and metabolic traits was examined using genetic correlations. Here, the GWAS of income was conditioned on a GWAS on intelligence using multi-trait-based conditional and joint analysis (mtCOJO). mtCOJO is used to perform conditional GWAS whereby the genetic effects from one GWAS are controlled for in another GWAS. Importantly, the mtCOJO method avoids well-known issues of collider bias that can occur by including heritable covariates[86]. In the current study, the GWAS on income was conditioned on a GWAS on intelligence (and the intelligence GWAS was

conditioned on the income GWAS) before the genetic correlations between income (and intelligence) and 27 variables mentioned above were reran.

**Genetic prediction.** The summary statistics from our GWAS of household income PGRSs were derived using PRSice-2[87] and the Generation Scotland:Scottish Family Health Study (GS:SFHS) cohort. The recruitment protocol and sample characteristics of GS:SFHS are described in full elsewhere[88,89]. In brief, 23,690 participants were recruited through their GP from across Scotland. Participants were all aged 18 years and over and were not ascertained based on the presence of any specific disease. Following the removal of individuals who preferred not to answer, income was assessed in GS:SFHS by 5-point scale (1 less than £10,000, 2 between £10,000 and £30,000, 3 between £30,000 and £50,000, 4 between £50,000 and £70,000 and 5 more than £70,000). Individuals who preferred not to answer were excluded from the analysis. Individuals who had taken part in UK Biobank were also removed from the GS:SFHS data set ($n = 174$). SNPs were included in the data if they had an MAF of ≥0.01 and Hardy–Weinberg $P$ value >0.000001. Finally, one from every pair of related individuals were removed from the data set by creating a genetic relationship matrix using GCTA[90] and removing individuals who are related at ≥0.025. This yielded a final sample size of 6680 participants who had genotype data and income data.

The participant's level of income was then used as a predictor in a regression analysis with age, sex and 20 principal components included to control for population stratification. The standardized residuals from this model were then used as each participant's income phenotype. PGRSs were created using the income phenotype derived using UK Biobank.

In each instance PRSice-2 was used to create five PGRSs corresponding to one of five P value cutoffs ($P \leq 0.01$, $P \leq 0.05$, $P \leq 0.1$, $P \leq 0.5$ and $P \leq 1$) applied to the association statistics from the summary data. The polygenic risk scores for each threshold were then standardized and used in a regression model to predict the income phenotype in GS:SFHS.

**Multi-trait analysis of GWAS.** MTAG[25] can be used to meta-analyze genetically correlated traits in order to increase power to detect loci in any one of the traits. Only summary data are required in order to carry out MTAG and bivariate LD score regression is carried out as part of an MTAG analysis to account for (possibly unknown) sample overlap between the GWAS data sets[25]. The goal of this analysis was to increase the power to detect loci associated with income, and so our income GWAS was meta-analyzed with the GWAS on years of education by Okbay et al.[91] using MTAG. Both the Okbay data set and the income data set from UK Biobank had a similar level of power (Okbay mean $\chi^2 = 1.65$, UK Biobank income mean $\chi^2 = 1.45$) and they showed a genetic correlation of $r_g = 0.77$ (SE = 0.02), confirming that both income and education, as measured using these data sets, have a highly similar genetic aetiology. Functional annotation and loci discovery, gene mapping, gene-based GWAS, gene set and gene-property analysis, were also performed using the MTAG-derived data set on income. In addition, following the removal of UK-based cohorts from the educational attainment summary statistics, genetic prediction was performed using the MTAG-derived income phenotype and the GS:SFHS as described above.

## Data availability

The household income association results, and the multivariate analysis conducted using MTAG can be downloaded from The Lothian Birth Cohorts of 1921 and 1936 data-sharing resource: https://www.lothianbirthcohort.ed.ac.uk/content/gwas-summary-data and at http://www.phenoscanner.medschl.cam.ac.uk/.

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

## Acknowledgements

This work was undertaken in The University of Edinburgh Centre for Cognitive Ageing and Cognitive Epidemiology (CCACE), supported by the cross-council Lifelong Health and Wellbeing initiative (MR/K026992/1). Generation Scotland received core support from the Chief Scientist Office of the Scottish Government Health Directorates [CZD/16/6] and the Scottish Funding Council [HR03006] and from the Medical Research Council UK and the Wellcome Trust (Wellcome Trust Strategic Award "STratifying Resilience and Depression Longitudinally" (STRADL) Reference 104036/Z/15/Z). Funding from the Biotechnology and Biological Sciences Research Council (BBSRC), the Medical Research Council (MRC) and the University of Edinburgh is gratefully acknowledged. CCACE funding supports IJD. C.H. was supported by an MRC University Unit Programme Grant MC_UU_00007/10 (QTL in Health and Disease). The Medical Research Council (MRC) and the University of Bristol support the MRC Integrative Epidemiology Unit [MC_UU_12013/1, MC_UU_12013/9 and MC_UU_00011/1]. The Economics and Social Research Council (ESRC) supports NMD via a Future Research Leaders grant [ES/N000757/1]. This work is part of a project entitled 'social and economic consequences of health: causal inference methods and longitudinal, intergenerational data', which is part of the Health Foundation's Efficiency Research Programme. Participants in INTERVAL were recruited with the active collaboration of NHS Blood and Transplant England (www.nhsbt.nhs.uk). DNA extraction and genotyping was co-funded by the National Institute for Health Research (NIHR), the NIHR BioResource (http://bioresource.nihr.ac.uk/)

and the NIHR [Cambridge Biomedical Research Centre at the Cambridge University Hospitals NHS Foundation Trust] [*]. The academic coordinating centre for INTERVAL was supported by core funding from NIHR Blood and Transplant Research Unit in Donor Health and Genomics (NIHR BTRU-2014-10024), UK Medical Research Council (MR/L003120/1), British Heart Foundation (RG/13/13/30194) and the NIHR [Cambridge Biomedical Research Centre at the Cambridge University Hospitals NHS Foundation Trust]. A complete list of the investigators and contributors to the INTERVAL trial is provided in ref. [16]. The academic coordinating centre would like to thank blood donor centre staff and blood donors for participating in the INTERVAL trial. *The views expressed are those of the authors and not necessarily those of the NHS, the NIHR or the Department of Health and Social Care. W.D.H. is supported by a grant from Age UK (Disconnected Mind Project). We thank George Davey Smith for his comments on an early version of this paper.

## Author contributions

W.D.H. conceptualization, data curation, data analysis, data visualization, interpretation of results, writing original draft, writing review and editing, project administration and co-ordination. N.M.D. data analysis, interpretation of results, writing review and editing. S.J.R. interpretation of results, writing review and editing. N.G.S. data analysis, interpretation of results, writing review and editing. J.B. writing review and editing. S.B. data analysis, interpretation of results, writing review and editing. E. Di A. writing review and editing. D.J.R. writing review and editing. S.X. data analysis, data visualization, writing review and editing. G.D. writing review and editing. D.C.M.L. writing review and editing. D.J.P. data curation, writing review and editing. C.H. data curation, writing review and editing. A.S.B. data curation, writing review and editing. A.M.M. writing review and editing. C.R.G. writing review and editing. I.J.D. funding acquisition, data interpretation, writing review and editing.

## Competing interests

The authors declare no competing interests.
