## [Peer Review File · Nature Communications]

Reviewers' Comments:

Reviewer #1:

Remarks to the Author:

Report for MS #NC-2019 GWAS of income

This is a nice paper. It updates the analysis of the initial release of the UK Biobank in Hill et al. (2016) to include the full release, finds many more results due to the increased statistical power, and includes many additional analyses.

These are my major comments:

1. In this context I don't believe the assumptions underlying Mendelian randomization or its various generalizations. For example, consider MR-Egger. It relies on the InSIDE (Instrument Strength Independent of Direct Effect) Assumption. If some SNPs are brain related, then those SNPs likely have bigger effects on both intelligence (the "instrument strength") and on personality traits that matter for income through other channels (the "direct effect"). Similar critiques apply to other generalizations of MR.

I strongly urge the authors to highlight the likely violations of the MR assumptions much more than they do, and to downplay these results much more than they do.

2. Because of the particular sensitivity of the topic – linking DNA with income differences across people, and examining intelligence as the mediator – and the potential for misinterpretation, I think it is even more important than usual for the paper to be accurate and careful in explaining how to interpret results. Unfortunately, I'm afraid the paper falls short in a number of places. Here are some examples:

A. Abstract: "These results are important for understanding the observed socioeconomic inequalities in Great Britain today." Although I think the authors merely meant that their results are related to socioeconomic differences across people (not a deep point, but stated in a way that sounds fitting for the end of an Abstract), a reader might easily interpret the sentence as suggesting that the paper shows that inequality is genetically determined (and, a reader might further reason erroneously, therefore nothing can be done about it). I believe the authors have a responsibility to avoid any statements that could be misunderstood that way.

B. p.28: "This finding indicates that genetic variants that are predictive of income might in part be so because they influence intelligence, and that it is intelligence that is an arguably more proximal causal factor, accounting for some of the differences in income in Great Britain today." The first part of this sentence gives too much credence to the MR results; it is plausible that the genetic variants operate through intelligence, but I don't think the MR results are very informative about that possibility. The second part again may be suggestive of genetic determinism and policy ineffectiveness. The same comments apply to the last sentence of the paper: "Furthermore, we identify intelligence as one of the causal psychological mechanisms partly driving differences in SEP in Great Britain today."

C. p.4: "The link between SEP and health is typically thought to be due to environmental factors including, but not limited to: access to resources, exposure to harmful or stressful environments, adverse health behaviors such as smoking, poor diet, and excessive alcohol consumption, and a lack of physical exercise. In addition, however, genetic factors have long been discussed as a partial explanation for the SEP-health association." This passage suggests that genetic factors are a distinct explanation from the environmental factors listed earlier, but that is not true. For example, genetic influences on what food people like could lead to poor diet (which in turn could lead to both low SEP and poor health).

In fact, while I believe that careful wording in the paper is essential, I would urge the authors to also do more than that. For example, they should consider writing an FAQ along the lines of those written by the SSGAC when studying sensitive topics like educational attainment (which is less sensitive than income and intelligence!).

3. The paper is somewhat repetitive, especially the Discussion section. I suggest looking for ways

to reduce repetitiveness throughout, but most importantly, focus the Discussion on new material, such as limitations, potential next steps, and big-picture issues. One idea, consistent with the previous point, would be to discuss appropriate interpretation of the results and potential misinterpretations.

4. To me, the most interesting results in the paper are the comparison between the genetic correlations of various phenotypes with income and their genetic correlations with educational attainment. I suggest highlighting these results more and speculating a bit more on the reasons why the genetic correlations are so similar for most phenotypes but quite different for a few.

5. I cannot find anywhere a statement of how the income variable was coded in the GWAS, but that needs to be clearly described. Was it just treated as a continuous 1-5 measure?! At a minimum, I'd want to see some robustness analyses to using some other reasonable coding scheme. Given that the GWAS is run on individual-level data in the UK Biobank, the correct approach would be ordinal regression (treating income as an ordered, categorical variable). I don't know if that's computationally feasible, but something along those lines should be doable.

Here are some typos I came across while reading the paper:

1. "High quality genotyping was performed on 332,050participants." – typo at end. Also, the sentence implies that the authors did the genotyping themselves, but I don't think that's the case.

2 p.8: should "MAF of 0.001" be "MAF >0.001"?

3. p.15: should "Hardy-Weinberg P-value of 0.000001" be "Hardy-Weinberg P-value > 0.000001"?

Reviewer #2:

Remarks to the Author:

This is an important paper that should be published eventually, but requires extensive revisions.

MAJOR POINTS

1. Mendelian randomization (MR) is not a reliable method for causal inference.

The authors do acknowledge the problem of "horizontal pleiotropy": SNPs that act as confounders of the two traits, thus producing a correlation between the two traits despite the possibility that neither trait affects the other. The authors try MR-Egger in an attempt to deal with this. It is well understood, however, that this attempted enhancement of MR will not work if there is a genetic correlation between the two traits ascribable to heritable confounding variables. A particularly clear demonstration of this reality was recently presented by O'Connor and Price (2018). Following a very reasonable simulation procedure, O'Connor and Price found that MR-Egger (and all other MR methods tested in their work) yield substantially elevated false-positive rates in the presence of a heritable confounder (see their Supplementary Table 2). In addition, these methods also yield substantially elevated false-positive rates if the two traits differ in polygenicity (i.e., number of causal SNPs) and GWAS sample size.

The authors might respond that because their MR-Egger intercept does not differ significantly from zero, they have evidence against horizontal pleiotropy. This test provides no such evidence, however, if the InSIDE assumption is violated. This is precisely what occurs "if the pleiotropic effects act via a confounder of the exposure-outcome association" (Bowden, Davey Smith, & Burgess, 2015, Box 1). In short, MR cannot be trusted and should be dropped entirely from the paper.

I think that the paper would suffice to meet the standards of publication in Nature Communications without any claims about directed causal relations between phenotypes. But I sense that the causal claims are important to the authors. This is why I recommend that they employ the Latent Causal Variable (LCV) method. The primary purpose of the O'Connor and Price paper mentioned above is not the investigation of MR methods (although the paper is valuable for this alone) but rather to present the LCV method, which in my view is more powerful than any MR method. Its false-positive rate remains well calibrated in a wide variety of situations, including many where MR methods fail; in non-null cases, where there is a genuine causal relationship, its estimate of the causal effect might conceivably be biased downward, but these are typically cases where MR methods perform poorly as well.

Sometimes my recommendations are not followed by authors or editors, and I imagine that this paper might ultimately be published with MR still in it. In this event, I ask that on p. 5 MR be described as an instrumental-variable technique rather than a method that exploits naturally occurring randomness. The term "Mendelian randomization" itself is a bit of a misnomer, since it is only within the same family that genotypes are randomly assigned. When we consider a population as a whole, it is a bit unclear whether we can analogize the assignment of genotype to a game of chance.

2. The manner in which the authors calculate the statistical significance of the difference between two genetic correlations, described in the caption of Fig. 5, is probably wrong. Their method would be sound if there were no sampling covariance between the two estimates. But in reality there almost certainly is sampling covariance, due to overlapping traits and samples. I would therefore calculate the SE of the difference using the block jackknife, as follows. Divide the genome into blocks of contiguous SNPs; different researchers have used different blocking schemes, and since I do not know of any one such scheme being superior to the others, I suggest a block length of 5 cM. Remove one block of SNPs, recalculate the two genetic correlations, and record their difference. Repeat for each block of SNPs. Finally, calculate the variance of the difference over blocks and multiply this by $(\# \text{ of blocks} - 1)(\# \text{ of blocks} + 1)$. This is the sampling variance, and the square root of it is the SE of the difference accounting for sampling covariance.

Some of the differences are extremely interesting. E.g., is intelligence more genetically correlated with autism than income? So getting the SE right is quite important.

3. The developmental analysis (the results of which are summarized in Supplementary Tables 12 and 13) is described far, far too sketchily. There is no relevant methods description anywhere in the paper, as far as I can tell. What exactly is the URL where the expression data can be obtained? The one given in the caption of Supplementary Table 12 has many further links. How were the data cleaned and processed? Are you sure that the proper reference for these data is Miller et al. (2014)? Although this is what it says on the Allen Institute website, I strongly suspect that this is a mistake and that the proper reference is Kang et al. (2011). If so, however, the stage definitions employed in the paper under review must be different from those used originally by Kang et al. All this needs to be clarified. How were genes assigned to the various stages? That is, what makes a gene belong to the "early prenatal" binary gene set rather than to "middle adulthood?" I direct the authors toward Pers et al. (2016) as a reasonably good model of a methods description.

Assuming that satisfactory answers can be given to all of these questions, I still perceive an inefficiency in the use that the authors make of the BrainSpan data. It seems to me that overall trends could be tested. For example, even if no single stage is statistically significant at the threshold 0.05/11, the difference between the average of the postnatal stages and the average of the prenatal stages might be significant. It seems safest to use the block jackknife to calculate the SE for such a test. This is only a suggestion; I am willing to sign off on the paper even if the authors do not go to such lengths. Clearing up the mysteries about the data source and so on, however, are absolutely essential.

4. My inclination is to omit the conditioning analysis whose results are presented in Supplementary Table 16. The authors use mtCOJO to estimate the associations of their SNPs with a phenotype (e.g., income) "conditioning on the genetic values of the covariate risk factors ... where the genetic value is defined as the aggregated effect of all SNPs on a phenotype accounting for LD" (Zhu et al., 2018, p. 10). I note that Zhu et al. call their method "approximate." Having (approximately) controlled for the genetic value of a covariate (e.g., intelligence) in this way, the authors then calculate the genetic correlation between adjusted income and some other trait (e.g., a health outcome). Now this method is *not* equivalent to regressing the genetic value of health on the genetic value of income while including the genetic value of intelligence as a covariate. Instead it is equivalent to calculating what is called the "semipartial" genetic correlation.

I object to this procedure because its result is hard to interpret. And besides the difficulty of interpreting semipartial correlations in general, we have the additional complication of these being only semipartial genetic correlations. Can we interpret the results as if we had performed them with the full phenotypic correlations?

I suspect that the authors will feel strongly about retaining these analyses. Here are some suggestions then for improvement, in order of my own preference.

(I) Calculate the partial genetic correlation. That is, partial out the covariate from both variables whose adjusted correlation is to be correlated. With the three zero-order genetic correlations, this can be easily be done by applying the standard formula (as given, e.g., by Wikipedia).

(II) Keep the semipartial genetic correlations but explain their meaning better. The authors do offer an interpretation on p. 29, but this is not particularly well done. I would employ an interpretation in terms of incremental R^2 in a regression with two predictors.

Regardless of what the authors do, it is absolutely essential that they calculate the SE of the difference between two correlations correctly, as described earlier.

I also suggest consulting the paper by Cesarini et al. (2016) about the effects of lottery wealth on various outcomes. Given the high quality of causal inference in this study, the authors would do well to check whether their suggestions and interpretations are consistent with its results.

5. The definition of MAGMA competitive testing on p. 10 is incorrect. The difference between self-contained and competitive testing is that the null hypothesis of the former posits no association between any set member and the phenotype; the null hypothesis of the latter is that the members of the set are no more associated with the phenotype than the background level (i.e., the partial regression coefficient of set membership in the prediction of gene-level significance is zero).

6. On p. 11, is "A minor allele frequency cut-off of >0.1 " a typo?

7. On p. 24, why is the SE of the LDSC intercept so large? This is baffling. And why was this analysis even done? If I was confident that neither trait's GWAS was subject to much confounding, I would use MTAG without too much concern.

MINOR

1. Why say "SEP" instead of the well-known "SES" or "social class?" Changing terminology and abbreviations to be confusing. At least once I had to go back and remind myself of what SEP means.

2. On p. 7, it says ">10 putative third degree relatives from the kinship table." What does this

mean?

3. I suggest either using an equation to give the definition of LD Score on p. 11 or not giving it at all. The present verbal definition is hard to understand.

4. In the heritability-partitioning analyses with stratified LD Score regression (s-LDSC), the authors use the original annotations supplied by Finucane et al. (2015). I recommend the use of the Gazal et al. (2017) annotations, which include the original s-LDSC annotations as a subset. The additional annotations refer to properties such as LD itself, MAF, allele age, and so forth. These are important to include because heritability per SNP varying by LD is a potentially major source of confounding in LD Score regression. I make this a minor point rather than a major one because my experience suggests that the rank orders of enrichments and so forth are similar with or without the Gazal et al. annotations; with the new annotations, the enrichment estimates tend to become closer to one.

5. The cell-type data used by Skene et al. (2018), which can be employed with MAGMA or s-LDSC, has been used in studies of three phenotypes to my knowledge: schizophrenia (Skene et al., 2018), neuroticism (Nagel et al., 2018), and intelligence (Savage et al., 2018) (which shows a strong genetic correlation with income). All three of these papers report significant or just-short-of-significant enrichment of medium spiny neurons and serotonergic neurons. It happens that the paper under review reports the same. I cannot point to any specific artifact in the data source or analysis, but I nevertheless find these results hard to accept. I suggest relegating these results to the supplement or inserting a caveat regarding the oddity of these a priori unlikely results being found regardless of the phenotype.

6. The authors use the original ten tissue annotations provided by Finucane et al. (2015) in their s-LDSC analyses. Each of these annotations, however, is formed by taking a union of SNPs associated with assayed histone marks across a variety of cell types and developmental stages. The SNPs that bear one of these annotations may be in fact quite heterogeneous. Therefore I suggest using the new and more specific tissue annotations provided by Finucane et al. (2018).

Also, even if the authors keep their analyses with the 2015 annotations, I object to this paragraph on p. 26: "Whereas the tissue of the central nervous system showed the highest level of enrichment, the adrenal/pancreas, skeletal muscle, cardiovascular, and immune/hematopoietic tissues all showed significant enrichment. The finding that the regions of the genome undergoing purifying selection, as well as tissue types from multiple biological systems are enriched in their associations with income, is consistent with the notion that, whereas intelligence differences might make some contributions to differences in income, a range of other partly-heritable phenotypes also likely to contribute." But the enrichment of other tissues may simply mean that SNPs associated with histone marks in one tissue may also be associated with such marks in another tissue. The authors need to look at the taus, that is, the partial regression coefficients of annotations.

7. The authors run s-LDSC on their income results that have been "enhanced" by MTAG. I regard the main uses of MTAG as identifying more genome-wide significant hits (and then performing downstream analyses of those hits) and obtaining a higher prediction R^2 with the PGS. I am skeptical of whole-genome analyses such as s-LDSC, where contamination of one phenotype by the other might be more of a concern. There needs to be more study of this issue. I approve of the authors relegating the s-LDSC results of MTAG-enhanced income to the supplement, but perhaps even there the authors should issue a caveat.

8. On p. 26, the authors say: "Medium spiny neurons have previously been linked to schizophrenia⁴² as well as to education.¹⁰" Reference 10 is Okbay et al. (2016), which did not use this method/dataset in the analysis of enriched tissues and cell types. So this thought is either wrong or incompletely expressed. In any event, if the authors follow one of my previous

suggestions, these findings will be relegated to the supplement.

9. On p. 27, the top paragraph is about evolutionarily conserved regions. But as far as I know, all phenotypes studied in GWAS show this enrichment. There is nothing special about income in this regard. It is thus a bit odd to make this a prominent finding for discussion.

10. On p. 27, the authors say: "Also consistent with the action of a mutation-selection balance was the observed correlation of 0.42 ($P=2.2 \times 10^{-4}$) between minor allele frequency and the effect size for the SNPs found in the independent genomic loci – this indicated that variants with a lower MAF have a greater association with income." This relationship is almost certainly real, but its magnitude will be exaggerated if there is no correction for statistical power. I suggest not ascertaining for significance and plotting the data in the manner of Yang et al. (2015)'s Fig. 4C and D. The method used to calculate the statistical significance of the trend in that paper is now admitted by the authors to be incorrect. If desired, statistical significance is likely to be calculated accurately by the block jackknife.

Bowden, J., Davey Smith, G., & Burgess, S. (2015). Mendelian randomization with invalid instruments: Effect estimation and bias detection through Egger regression. *International Journal of Epidemiology*, 44, 512-525.

Cesarini, D. et al. (2016). Wealth, health, and child development: Evidence from administrative data on Swedish lottery players. *Quarterly Journal of Economics*, 131, 687-738.

Finucane, H. K. et al. (2015). Partitioning heritability by functional annotation using genome-wide association summary statistics. *Nature Genetics*, 47, 1228-1235.

Finucane, H. K. et al. (2018). Heritability enrichment of specifically expressed genes identifies disease-relevant tissues and cell types. *Nature Genetics*, 50, 621-629.

Gazal, S. et al. (2017). Linkage disequilibrium-dependent architecture of human complex traits shows action of negative selection. *Nature Genetics*, 49, 1421-1427.

Kang, H. J. et al. (2011). Spatio-temporal transcriptome of the human brain. *Nature*, 478, 483-489.

Miller, J. A. et al. (2014). Transcriptional landscape of the prenatal human brain. *Nature*, 508, 199-206.

Nagel, M. et al. (2018). Meta-analysis of genome-wide association studies in 449,484 individuals identifies novel genetic loci and pathways. *Nature Genetics*, 50, 920-927.

O'Connor, L. J. & Price, A. L. (2018). Distinguishing genetic correlation from causation across 52 diseases and complex traits. *Nature Genetics*, 50, 1728-1734.

Pers, T. H. et al. (2016). Comprehensive analysis of schizophrenia-associated loci highlights ion channel pathways and biologically plausible candidate causal genes. *Human Molecular Genetics*, 25, 1247-1254.

Savage, J. E. et al. (2018). Genome-wide association meta-analysis in 269,867 individuals identifies new genetic and functional links to intelligence. *Nature Genetics*, 50, 912-919.

Skene, N. G. et al. (2018). Genetic identification of brain cell types underlying schizophrenia. *Nature Genetics*, 50, 825-833.

Yang, J. et al. (2015). Genetic variance estimation with imputed variants finds negligible missing heritability for human height and body mass index. *Nature Genetics*, 47, 1114-1120.

Zhu, Z. et al. (2018). Causal associations between risk factors and common diseases inferred from GWAS summary data. *Nature Communications*, 9, 224.

Reviewer #1

Comment: “This is a nice paper. It updates the analysis of the initial release of the UK Biobank in Hill et al. (2016) to include the full release, finds many more results due to the increased statistical power, and includes many additional analyses.”

Response: We are grateful for this positive evaluation.

Comment: “In this context I don’t believe the assumptions underlying Mendelian randomization or its various generalizations. For example, consider MR-Egger. It relies on the InSIDE (Instrument Strength Independent of Direct Effect) Assumption. If some SNPs are brain related, then those SNPs likely have bigger effects on both intelligence (the “instrument strength”) and on personality traits that matter for income through other channels (the “direct effect”). Similar critiques apply to other generalizations of MR.

I strongly urge the authors to highlight the likely violations of the MR assumptions much more than they do, and to downplay these results much more than they do.”

Response: Obtaining valid Mendelian randomization estimates depends on satisfying the three assumptions that define an instrumental variable. First, the SNPs must associate with intelligence; second, the SNPs must only affect income via their effect on intelligence; and third, there must be no common causes of both the SNPs and income.

It is not possible to definitively prove that the second or third assumptions holds. However, to violate the assumptions of the MR analysis, it is not sufficient for SNPs to have pleiotropic effects on other phenotypes. Instead, these effects must be horizontally (not vertically) pleiotropic and directly affect the outcome via mechanisms other than intelligence. Vertically pleiotropic effects occur when the effect of the SNPs on the outcome (income) are mediated first by the exposure of interest (intelligence), then another trait (e.g. conscientiousness). Or vice versa, when the effects of SNPs on the outcome (income) are mediated first by another trait (e.g. brain volume), which affect intelligence. Horizontally pleiotropic on the other hand occur when the SNPs affect other traits and these traits affect income without being mediated via intelligence. For example, if a SNP directly affects both intelligence and conscientiousness, and conscientiousness affects income. This excludes many potential pleiotropic effects, such as a vertically pleiotropic causal chain from a SNP to increased brain volume, to intelligence, to income. Similarly, vertically pleiotropic effects that are downstream of intelligence also will not cause bias in our estimated causal effect of

intelligence on income (For example, a causal chain from SNP to intelligence, to education, to income). The SNPs identified in the intelligence GWAS are also associated with some personality traits or brain related phenotypes – (for example neuroticism which is genetically correlated with both intelligence ($r_g = -0.23$, $SE = 0.02$, $P = 1.83 \times 10^{-23}$) and with income ($r_g = -0.39$, $SE = 0.04$, $P = 8.56 \times 10^{-24}$).¹ We also investigated the robustness of our results using the MR-Egger estimator; this provides an unbiased estimate of the causal effect of intelligence on income if the pleiotropic effects on other traits are independent of the effects of the SNPs on intelligence.² Other estimators are robust to other specific forms of pleiotropy, for example the weighted median and weighted mode; however, we cannot prove that the assumptions underlying any of these estimators are true. However, our estimates of the effects of intelligence are consistent across a range of estimators which depend on related, but distinct assumptions. Therefore, we draw our conclusions by triangulating our conclusions across multiple models, which is likely to increase the reliability of our inferences.

Furthermore, it is possible (indeed almost certain) that a trait closely related to intelligence is the underlying causal factor. Trivially, this can be seen because our measure of intelligence are scores on a range of cognitive ability tests. The UK Biobank, where the bulk of our participants for our intelligence GWAS came from, measured income prior to conducting the fluid intelligence test. Therefore, it is temporally impossible for the score on the test to affect household income. Thus, our results clearly relate to the underlying (latent) trait – intelligence, rather than the scores on the test. It should be noted, though, that whether the true underlying causal trait is solely “intelligence”, or a mixture of related cognitive and personality traits (e.g. neuroticism) is currently unclear. As we obtain more data for more detailed intelligence and personality phenotypes, we will be able to use other methods, such as multivariable Mendelian randomization,^{3,4} to investigate the true underlying causal mechanism.

However, we do agree with the reviewer that it is important to be more explicit with potential violations of the assumptions of MR and to downplay the results more than we did in the initial submission. In response to the reviewer’s comment, then, we have added these important caveats to the MR analysis to our manuscript. The revised text is highlighted below in bold to more easily draw attention to these additions.

As asked for by the reviewer we downplay the results of the MR analysis on the pages highlighted below.

On page 5

“This random allocation of intelligence at conception can be used to indicate the causal relationship between intelligence and income, **in situations where the assumptions of MR are met.**”

On page 23

“**Should the assumptions of the MR hold,** this indicates that greater intelligence causes a higher level of income.”

On page 32

“This finding indicates that, **if the assumptions of MR in this instance are met,** genetic variants that are predictive of income might in part be so because they influence intelligence, and that it is intelligence that is an arguably more proximal causal factor, accounting for some of the differences in income in Great Britain today.”

We also highlight where the assumptions of MR may be violated by dynastic effects. This can be found on page 36.

“**A limitation of the Mendelian Randomization analysis specifically are potential dynastic effects, which may violate the assumptions of MR. Dynastic effects are where genetic variants that the parent has but the child does not, are associated with parental behaviors, and these parenting behaviours are a causal factor in the SEP of the child.**⁵

An example of this would be that parents with a greater predisposition towards intelligence are also those that are more likely to provide opportunities for their children to enter higher-income occupations. **In this instance the second assumption of Mendelian Randomization, that the instrument must only affect income via their effect on intelligence, would be violated as the association of the offspring’s SNPs and income would be partially due to the effects we observe may be due to the of the parents’ genotype effects on their parents’ intelligence which subsequently affects offspring income.** Whereas the current data cannot differentiate between causality and dynastic effects, it should be noted that, for another measure of SEP, educational attainment, there is evidence of indirect genetic effects which account for ~30% of the variance of the direct genetic associations.⁶ Future work in multi-generational samples should examine the role that such indirect genetic effects play in individual differences in income, as well as if their presence (if established) could result in an inflation of the estimate for a causal effect using Mendelian Randomization.⁷”

Comment: “Because of the particular sensitivity of the topic – linking DNA with income differences across people, and examining intelligence as the mediator – and the potential for misinterpretation, I

think it is even more important than usual for the paper to be accurate and careful in explaining how to interpret results. Unfortunately, I'm afraid the paper falls short in a number of places. Here are some examples:

A. Abstract: "These results are important for understanding the observed socioeconomic inequalities in Great Britain today." Although I think the authors merely meant that their results are related to socioeconomic differences across people (not a deep point, but stated in a way that sounds fitting for the end of an Abstract), a reader might easily interpret the sentence as suggesting that the paper shows that inequality is genetically determined (and, a reader might further reason erroneously, therefore nothing can be done about it). I believe the authors have a responsibility to avoid any statements that could be misunderstood that way.

B. p.28: "This finding indicates that genetic variants that are predictive of income might in part be so because they influence intelligence, and that it is intelligence that is an arguably more proximal causal factor, accounting for some of the differences in income in Great Britain today." The first part of this sentence gives too much credence to the MR results; it is plausible that the genetic variants operate through intelligence, but I don't think the MR results are very informative about that possibility. The second part again may be suggestive of genetic determinism and policy ineffectiveness. The same comments apply to the last sentence of the paper: "Furthermore, we identify intelligence as one of the causal psychological mechanisms partly driving differences in SEP in Great Britain today."

C. p.4: "The link between SEP and health is typically thought to be due to environmental factors including, but not limited to: access to resources, exposure to harmful or stressful environments, adverse health behaviors such as smoking, poor diet, and excessive alcohol consumption, and a lack of physical exercise. In addition, however, genetic factors have long been discussed as a partial explanation for the SEP-health association." This passage suggests that genetic factors are a distinct explanation from the environmental factors listed earlier, but that is not true. For example, genetic influences on what food people like could lead to poor diet (which in turn could lead to both low SEP and poor health).

In fact, while I believe that careful wording in the paper is essential, I would urge the authors to also do more than that. For example, they should consider writing an FAQ along the lines of those written by the SSGAC when studying sensitive topics like educational attainment (which is less sensitive than income and intelligence!)."

Response: We agree with the reviewer that this phenotype should be handled in a sensitive manner given the potential for misinterpretation by academics, the media and general public. As the reviewer suggests, we have now included an FAQ document. This is first referenced in the discussion section on page 36 where we state that

“A further limitation is that molecular genetic analyses of phenotypes such as intelligence, income, or SEP, appear prone to being misinterpreted.⁸ Such misunderstandings include describing associated variants as “genes for income”, or the misinterpretation that any associated variant, and indeed any non-zero heritability estimate, is evidence for genetic determinism or the immutable nature of these phenotypes via environmental intervention. Such interpretations are incorrect because, as indicated by the heritability estimate derived in the current study, the vast majority of differences in income in Great Britain today are associated with environmental differences between individuals rather than any genetic differences. We include a figure (Figure 1) that illustrates that genetic variants do not act directly on income; instead, genetic variants are associated with partly heritable traits (such as intelligence, conscientiousness, health etc.) which have their own complex gene-to-phenotype paths (including neural variables) and are ultimately associated with income. Therefore, the genetic variant-income associations discovered here are no more “for” income than they are “for” these other traits. For more discussion of the implications of these results, aimed at the general reader, we have provided a “Frequently Asked Questions” (FAQ) document in the Supplementary Materials.”

We have now modified some of the language to stress the “non-deterministic” nature of our findings. In the first instance, in the abstract, we now write

“These results contribute toward understanding some of the observed socioeconomic inequalities in Great Britain today.”

We think that it is important to retain the qualifying statement of “in Great Britain today” because genetic associations made with measures of SEP are very likely to be different across cultures, as well as different within the same culture at different points in time. We expand on these caveats in the discussion section on page 37.

In the second instance, the reviewer is correct that this statement is dependent on the assumptions of Mendelian Randomisation being met. This sentence has now been amended to include this dependency. The full sentence now reads.

“This finding indicates that, if the assumptions of MR in this instance are met, genetic variants that are predictive of income might in part be so because they influence intelligence, and that it is intelligence that is an arguably more proximal causal factor, accounting for some of the differences in income in Great Britain today.”

In the third instance, the reviewer is again correct. We have now included the example provided by the reviewer to call attention to the idea that genetic factors are not distinct from environmental factors. This sentence now reads

“In addition, however, genetic factors have long been discussed as a partial explanation for the SEP-health association; for example, genetic predispositions towards certain diseases, and/or genetic influences on what foods people like, could lead to poor diet which in turn could lead to both lower SEP and poorer health.”⁹”

Comment: “The paper is somewhat repetitive, especially the Discussion section. I suggest looking for ways to reduce repetitiveness throughout, but most importantly, focus the Discussion on new material, such as limitations, potential next steps, and big-picture issues. One idea, consistent with the previous point, would be to discuss appropriate interpretation of the results and potential misinterpretations.”

Response: We have now amended the manuscript to ensure that there is less repetition. Furthermore, we have included some of the possible misinterpretations of our results on page 37, as described in the response to the Reviewer#1 in the above comment.

Comment: “To me, the most interesting results in the paper are the comparison between the genetic correlations of various phenotypes with income and their genetic correlations with educational attainment. I suggest highlighting these results more and speculating a bit more on the reasons why the genetic correlations are so similar for most phenotypes but quite different for a few.”

Response: We agree with the Reviewer that these are especially interesting results. We have now expanded our discussion section with further attempts to explain these findings, and we have re-designed the figures (now **Figures 6A, B, and C**) to highlight this contrast better. This can be found on page 32 of the revised manuscript, and we have copied in the additional text below for convenience.

“Fifth, we provide the best-to-date estimates of genetic correlations with income. This includes a genetic correlation with longevity ($r_g=0.47$, $SE=0.07$, $P=1.29\times 10^{-10}$). More importantly, this paper used genetic correlations to determine if the genetic contributions to income were significantly different in relation to health than another measure of SEP, years of education.

Our genetic correlations show that income and education each have similar genetic correlations with many variables. However, some genetic correlations differ depending on whether income or education is used as a measure of SEP, and those that differ tend to be those related to mental health. In those, the income genetic correlations show a direction of effect that is more beneficial to psychological health (**Figure 6A & Supplementary Table 21**). These results indicate that, genetic variants related to mental health might have more relevance for income differences than for education differences. This could be a life course phenomenon, i.e. education tends to be completed earlier in the life course, before some illnesses appear that could affect earning capability.

It should also be considered that these significantly different genetic correlations between education and income indicate that educational attainment serves to provide access to opportunities in the labour market, and those that have these opportunities are then better placed to engage in health-relevant behaviours. This would indicate that, whereas income may be a more distal phenotype from DNA than education, it is closer to outcomes such as later-life health, as shown by the significantly different genetic correlations. Future work should examine models where DNA > neuronal properties -> intelligence -> education -> income -> health, using multivariable Mendelian randomization^{10, 11, 12} to gauge the direct and indirect effects of income and education on health outcomes.

However, previous work using lotteries in Sweden as natural experiments to examine the causal effect of wealth on health differences¹³ found that, in the 10 years after receiving a prize (either as a single payment or multiple instalments), winning participants did not have a longer life or fewer hospital admissions compared with those who did not win the lottery.¹³ This indicates that, whereas high earners may be in better health and have a greater level of education than low earners, a high income might not be causal in such differences in affluent countries that have strong social support systems. Furthermore, children born to lottery winners were not found to be advantaged in terms of their level of scholastic performance compared to the children of those who did not win the lottery,¹³ a finding that argues against a dynastic effect mediated via wealth. Together, Whereas any causal effects of wealth on health are likely to differ across countries and times, should the results of this Swedish study generalize to the UK today, they would complement our results and together would support a model whereby genetic differences that are linked with health might be linked to partly heritable intermediary phenotypes, such as intelligence, that occur between genetic inheritance and differences in household income.

In this scenario, the similarities and differences between the genetic correlations derived using education and income might be explained in part by the differences in the intermediary

phenotypes that give rise to each of these measures of SEP. Under this model, the observed differences between genetic correlations with mental health (**Figure 6A**) would be due to intermediary variables that make a greater contribution to both income and mental health than they do to education. The commonalities observed between the genetic correlations derived using education and income with health would be indicate a similar contribution from intermediary phenotypes to income, education, and health.”

Comment: “I cannot find anywhere a statement of how the income variable was coded in the GWAS, but that needs to be clearly described. Was it just treated as a continuous 1-5 measure?! At a minimum, I’d want to see some robustness analyses to using some other reasonable coding scheme. Given that the GWAS is run on individual-level data in the UK Biobank, the correct approach would be ordinal regression (treating income as an ordered, categorical variable). I don’t know if that’s computationally feasible, but something along those lines should be doable.”

Response: The description of how the income variable was coded can be found on page 6 under the section titled “phenotype description”. The reviewer is correct in that we treated the 1 – 5 as a continuous measure in a linear regression. We also state on page 7 that

“The level of household income as measured on the 5 point scale was subjected to a regression using income as the outcome as has been conducted previously,¹⁴ and 40 genetic principal components (to control for population stratification), genotyping array, batch, age, and sex as predictors. The residuals from this model were then used in a GWAS assuming an additive genetic model as implemented in BGENIE.¹⁵”

This was done as there were no methods at the time of analysis that allowed ordinal regression of GWAS data to take place. Furthermore, previous work looking at household income has also treated ordinal data as continuous¹⁴ and found similar results to our own with strong genetic correlations with individual measures of SEP, notably educational achievement using the number of years of education a participant has ($r_g = 0.73$).

It should also be noted that, by measuring education as a binary variable based on achievement of a college or university level degree and then treating this case control data as continuous,^{16, 17} results in a phenotype genetically indistinguishable (the genetic correlation between the two variables was 1.¹⁸) with the number of years of education a participant had completed which is a continuous score. This example from the literature shows that treating a

case control variable as a continuous variable produces an almost identical phenotype when compared to a truly continuous measure of the same trait.

Comment: Here are some typos I came across while reading the paper:

1. “High quality genotyping was performed on 332,050 participants.” – typo at end. Also, the sentence implies that the authors did the genotyping themselves, but I don’t think that’s the case.

2 p.8: should “MAF of 0.001” be “MAF >0.001”?

3. p.15: should “Hardy-Weinberg P-value of 0.000001” be “Hardy-Weinberg P-value > 0.000001”?

Response: We have now corrected these typos – we thank the reviewer for noticing them. For the first sentence, added a qualifier to explain that UK Biobank performed the genotyping.

Reviewer #2

Comment: Mendelian randomization (MR) is not a reliable method for causal inference.

The authors do acknowledge the problem of "horizontal pleiotropy": SNPs that act as confounders of the two traits, thus producing a correlation between the two traits despite the possibility that neither trait affects the other. The authors try MR-Egger in an attempt to deal with this. It is well understood, however, that this attempted enhancement of MR will not work if there is a genetic correlation between the two traits ascribable to heritable confounding variables. A particularly clear demonstration of this reality was recently presented by O'Connor and Price (2018). Following a very reasonable simulation procedure, O'Connor and Price found that MR-Egger (and all other MR methods tested in their work) yield substantially elevated false-positive rates in the presence of a heritable confounder (see their Supplementary Table 2). In addition, these methods also yield substantially elevated false-positive rates if the two traits differ in polygenicity (i.e., number of causal SNPs) and GWAS sample size.

The authors might respond that because their MR-Egger intercept does not differ significantly from zero, they have evidence against horizontal pleiotropy. This test provides no such evidence, however, if the InSIDE assumption is violated. This is precisely what occurs "if the pleiotropic effects act via a confounder of the exposure-outcome association" (Bowden, Davey Smith, & Burgess, 2015, Box 1). In short, MR cannot be trusted and should be dropped entirely from the paper.

I think that the paper would suffice to meet the standards of publication in Nature Communications without any claims about directed causal relations between phenotypes. But I sense that the causal claims are important to the authors. This is why I recommend that they employ the Latent Causal Variable (LCV) method. The primary purpose of the O'Connor and Price paper mentioned above is not the investigation of MR methods (although the paper is valuable for this alone) but rather to present the LCV method, which in my view is more powerful than any MR method. Its false-positive rate remains well calibrated in a wide variety of situations, including many where MR methods fail; in non-null cases, where there is a genuine causal relationship, its estimate of the causal effect might conceivably be biased downward, but these are typically cases where MR methods perform poorly as well.

Response: Whereas we disagree with the reviewer that the MR results should be discarded from the paper, we do agree that they should be more carefully caveated as also suggested by reviewer #1. The changes we've made to the manuscript, both to downplay the MR results

and to show situations in which the assumptions of MR maybe violated, are found in response to comment one of reviewer one above.

Furthermore, the Latent Causal Variable (LCV) method suggested by reviewer two is an interesting potential Mendelian randomization estimator. Rather than estimating the causal effect of one phenotype on another, it estimates a “genetic causality proportion” (GCP) using SNP-phenotype associations across the entire genome, rather than a selected set of SNPs used in other Mendelian randomization estimators. The approach presumes that there is a latent variable that affects the putative exposure and outcome. They argue, as in other bidirectional Mendelian randomization approaches, that if the putative exposure affects the putative outcome, then SNPs strongly associated with the exposure should have large effects on the outcome, but not vice versa. They propose an estimator which exploits the fourth moment (kurtosis) of the SNP-phenotype associations (i.e. $E(\alpha_1^2\alpha_1\alpha_2)$ and $E(\alpha_2^2\alpha_1\alpha_2)$).

O’Connor and Price (2018)¹⁹ present a SNP level simulation in which a proportion of the SNPs have direct effects on the exposure, a proportion have direct effects on the outcome, and some affect both traits via the latent variable (i.e. horizontal pleiotropy). They present results from the simulations indicating false positive rates. The simulations presented in O’Connor and Price are highly favourable to LCV, however, it is unclear how closely the simulations reflect the relationship between intelligence and income. Specifically, what proportion of SNPs have a) effects on intelligence, and subsequently income, b) effects on income, and subsequently intelligence, c) horizontally pleiotropic effects on both income and intelligence, and d) It is not clear what the 4th moment (kurtosis) of the SNP-intelligence associations are, and whether this is sufficient to estimate the effects of intelligence on income. In future work, we may be able to investigate whether it is intelligence or some other closely related phenotype that causes these results. See response above to reviewer 1.

Recent papers, such as Howey et al. (2019)²⁰ have evaluated LCV using an independent simulation. They found some limitations in the approach, particularly with respect to power and performance under a null (no causal effects of the exposure on the outcome) simulation, writing:

“For LCV, the results are rather poor, presumably as the method is primarily designed to detect a genetic causality proportion (GCP) (which is not directly encapsulated by our simulation model), and genetic confounding effects are often problematic when not accounted for. There is a very low detection rate for the GCP when there is a causal relationship between the metabolite and Y in either direction. Somewhat perversely, the detection rate for a direct

causal relationship, in either direction, is much higher when there are no effects (Fig 10, left hand panels) compared”

O’Connor and Price (2018) applied their technique to a large number of relationships (Supplementary Table 12). These estimates recapitulate many previously reported causal effects, for example detecting an effect an effect of raised BMI on risk of heart attacks. LCV does not detect established causal effects (e.g. the effect of BMI on type 2 diabetes). LCV also finds strong evidence other putative causal relationships, such as an effect of hbA1c on BMI ($P=1.30\times 10^{-16}$), LDL on attending college ($P=2.8\times 10^{-10}$), an effect of high cholesterol on smoking status ($P=1.1\times 10^{-18}$) and neuroticism ($P=4.6\times 10^{-14}$). The IWV estimator found little evidence that these genetic correlations were causal. While these relationships are theoretically possible, we need more information about the reliability of these estimators before using LCV more widely in empirical studies.

Regarding “In addition, these methods also yield substantially elevated false-positive rates if the two traits differ in polygenicity (i.e., number of causal SNPs) and GWAS sample size.” We are not aware of any study (simulated or otherwise) that demonstrates inflated false positive rates if there are different number of causal SNPs for the two traits, if the Mendelian randomization assumptions hold. In specific situations, such as if a certain proportion of the SNPs are presumed to have horizontally pleiotropic effects, such as in Fig 2c in O’Connor and Price (2018), then Mendelian randomization methods are likely to have elevated false positive rates. However, it is unclear whether these simulations are a realistic approximation for the effects of intelligence on income, both in terms of strength of effects, frequency of pleiotropic and non-pleiotropic variants.

In summary, LCV is an interesting technique with potential, but at this point its relative strengths and limitations have not been established in the literature. Our current paper is an empirical rather than methodological contribution, so we are not in a position to undertake a thorough review of LCV and its assumptions needed for us to apply it in our manuscript.

Reviewer: *The manner in which the authors calculate the statistical significance of the difference between two genetic correlations, described in the caption of Fig. 5, is probably wrong. Their method would be sound if there were no sampling covariance between the two estimates. But in reality there almost certainly is sampling covariance, due to overlapping traits and samples. I would therefore calculate the SE of the difference using the block jackknife, as follows. Divide the genome into blocks of contiguous SNPs; different researchers have used different blocking schemes, and since I do not*

*know of any one such scheme being superior to the others, I suggest a block length of 5 cM. Remove one block of SNPs, recalculate the two genetic correlations, and record their difference. Repeat for each block of SNPs. Finally, calculate the variance of the difference over blocks and multiply this by $(\# \text{ of blocks} - 1) * (\# \text{ of blocks} + 1)$. This is the sampling variance, and the square root of it is the SE of the difference accounting for sampling covariance.*

Some of the differences are extremely interesting. E.g., is intelligence more genetically correlated with autism than income? So getting the SE right is quite important.

Response: The approach the reviewer suggests would be very difficult to achieve computationally as it would require performing many thousands of genetic correlations and storing a very large number of data sets. Specifically, we would need to store as many thousands of GWAS data sets, each with 5 Cm blocks extracted as the number of block jackknives required. Additionally, this would have to be done for each of our 29 phenotypes.

The method we have used here has also been used before^{21, 22, 23} (as this is simply from the formula for a sum or difference of normal distributions) whereas the method proposed by the reviewer has not been peer reviewed. We judge that this standard method is appropriate, as it is more conservative in the event that there is the sampling covariance described by the reviewer. This would mean that the differences we have identified are very unlikely to be due to a type I error.

We have now re-designed **Figure 6** for readers to see these differences between genetic correlations. In it, we see that the only instance where two sets of error bars do not overlap and our method states that the difference between the genetic correlations is not significant is for the MDD phenotype in **Figure 6A**.

Reviewer: *The developmental analysis (the results of which are summarized in Supplementary Tables 12 and 13) is described far, far too sketchily. There is no relevant methods description anywhere in the paper, as far as I can tell. What exactly is the URL where the expression data can be obtained? The one given in the caption of Supplementary Table 12 has many further links. How were the data cleaned and processed? Are you sure that the proper reference for these data is Miller et al. (2014)? Although this is what it says on the Allen Institute website, I strongly suspect that this is a mistake and that the proper reference is Kang et al. (2011). If so, however, the stage definitions employed in the paper under review must be different from those used originally by Kang et al. All this needs to be clarified. How were genes assigned to the various stages? That is, what makes a gene belong to the*

"early prenatal" binary gene set rather than to "middle adulthood?" I direct the authors toward Pers et al. (2016) as a reasonably good model of a methods description.

Response: We agree with the reviewer and have now updated the section on page 10 that describes this analysis. Furthermore, the reviewer is correct that the reference is indeed Kang et al (2011) and not Miller et al. (2011) as we stated. We have now replaced the Miller reference with the Kang et al reference. In addition, the revised method section on page 11 now reads as follows.

“An additional gene property analysis was conducted to determine if transcription in the brain at 11 developmental stages,²⁴ or across 29 different age groupings,²⁴ was associated with a gene’s link to household income. These RNA-Seq GEncode v10 summarized to genes data were accessed using the following link

http://www.brainspan.org/api/v2/well_known_file_download/267666525. The detailed descriptions of the normalization processes used can be found in the technical white paper at <http://help.brain->

[map.org/download/attachments/3506181/Transcriptome_Profiling.pdf?version=1&modificationDate=1382036562736&api=v2](http://help.brain-map.org/download/attachments/3506181/Transcriptome_Profiling.pdf?version=1&modificationDate=1382036562736&api=v2), where a total of 524 samples were available for analysis.

The developmental stages were assigned to each two groups (11 developmental stages and 29 age groupings) based on the age of the sample. The groupings of 25 post-conception weeks (pcw) and 35 pcw were excluded from the age groups as they contained fewer than three samples. Next, the 52,376 annotated genes were filtered so that the average Reads Per Kilobase (RPKM) is >1. This was performed in the developmental group and in the age group separately. This resulted in 19,601 genes for the developmental stage groupings and 21,001 genes for the age groupings. RPKM was then winsorized at 50 (RPKM>50 was replaced with 50). Then, the average of log transformed RPKM with a pseudocount 1 ($\log_2(\text{RPKM}+1)$) per group (for either 11 developmental stages or 29 age groups) was used as a covariate conditioning on the average across all the labels. To control for multiple tests a Bonferroni correction was used to control for 11 and 29 tests separately.”

Comment: My inclination is to omit the conditioning analysis whose results are presented in Supplementary Table 16. The authors use mtCOJO to estimate the associations of their SNPs with a phenotype (e.g., income) "conditioning on the genetic values of the covariate risk factors ... where the genetic value is defined as the aggregated effect of all SNPs on a phenotype accounting for LD" (Zhu et al., 2018, p. 10). I note that Zhu et al. call their method "approximate." Having (approximately) controlled for the genetic value of a covariate (e.g., intelligence) in this way, the authors then calculate the genetic correlation between adjusted income and some other trait (e.g., a health outcome). Now this method is **not** equivalent to regressing the genetic value of health on the genetic value of income while including the genetic value of intelligence as a covariate. Instead it is equivalent to calculating what is called the "semipartial" genetic correlation.

I object to this procedure because its result is hard to interpret. And besides the difficulty of interpreting semipartial correlations in general, we have the additional complication of these being only semipartial genetic correlations. Can we interpret the results as if we had performed them with the full phenotypic correlations?

Response: We disagree with reviewer two's interpretation of the mtCOJO method. This method has been used by the authors of the methods manuscript to perform an analysis with the same goals as those outlined here but with the focus on psychiatric rather than SEP variables.²⁵ The authors use the mtCOJO method to "adjust GWAS summary statistics of one disorder for the effects of genetically correlated traits to identify putative disorder-specific SNP associations." And, they also used it to "identify SNPs with disorder-specific effects by using independently collected genome-wide association study (GWAS) summary statistics for different disorders (thereby maximising contributing sample sizes)." The reviewer is correct, however, regarding the standard errors of the genetic correlations and we have now implemented this suggestion.

Reviewer: *I also suggest consulting the paper by Cesarini et al. (2016) about the effects of lottery wealth on various outcomes. Given the high quality of causal inference in this study, the authors would do well to check whether their suggestions and interpretations are consistent with its results.*

Response: We agree with the reviewer that this paper should be included in our discussion of the link between wealth and health. We have now added the paragraph below to our manuscript. It can be found on page 33 but is copied in below in full.

“However, previous work using lotteries in Sweden as natural experiments to examine the causal effect of wealth on health differences¹³ found that, in the 10 years after receiving a prize (either as a single payment or multiple instalments), winning participants did not display a longer life or fewer hospital admissions compared with those who did not win the lottery.¹³ Furthermore, children born to lottery winners were not advantaged in terms of their level of scholastic performance compared to the children of those who did not win the lottery.¹³ Together, this indicates that, whereas high earners may be in better health and have a greater level of education than low earners, a high income might not be causal in such differences in affluent countries that have strong social support systems. Whereas any causal effects of wealth on health are likely to differ across countries and times, should the results of this Swedish study generalize to the UK today, they would complement our results and together would support a model whereby genetic differences that are linked with health might be linked to partly heritable intermediary phenotypes, such as intelligence, that occur between genetic inheritance and differences in household income.

In this scenario, the similarities and differences between the genetic correlations derived using education and income might be explained in part by the differences in the intermediary phenotypes that give rise to each of these measures of SEP. Under this model, the observed differences between genetic correlations with mental health (**Figure 6A**) would be due to intermediary variables that make a greater contribution to both income and mental health than they do to education. The commonalities observed between the genetic correlations derived using education and income with health would be indicate a similar contribution from intermediary phenotypes to income, education, and health.”

Comment: “The definition of MAGMA competitive testing on p. 10 is incorrect. The difference between self-contained and competitive testing is that the null hypothesis of the former posits no association between any set member and the phenotype; the null hypothesis of the latter is that the members of the set are no more associated with the phenotype than the background level (i.e., the partial regression coefficient of set membership in the prediction of gene-level significance is zero).”

Response: We disagree with Reviewer #2 on this point. The background level of enrichment that the Reviewer talks about is the level of enrichment found in genes outside the gene set. In our manuscript on page 10 we state that

“Competitive testing, conducted in MAGMA,²⁶ examines if genes within the gene-set are more strongly associated with the trait of interest than other genes...”

When looking at the paper detailing the MAGMA method²⁶ in the introduction on page 2/19 it states that

“and the competitive type testing whether the association in the gene set is greater than in other genes”

And in the section titled “Gene-set analysis” in the MAGMA method paper on page 4/19 it states that

“Competitive gene-set analysis tests whether the genes in a gene-set are more strongly associated with the phenotype of interest than other genes.”

Comment: On p. 11, is "A minor allele frequency cut-off of >0.1" a typo?

Response: This is not a typo and these are the standard MAF and INFO score cut offs used for LDSC regression.^{18,27} This sentence has been made clearer –we now state that “SNPs were included if they had a minor allele frequency of > 0.1 and an imputation quality score of > 0.9.”

Comment: “On p. 24, why is the SE of the LDSC intercept so large? This is baffling. And why was this analysis even done? If I was confident that neither trait's GWAS was subject to much confounding, I would use MTAG without too much concern.”

Response: We thank the reviewer for their attention to detail here. This figure was misreported and should have stated SE = 0.0098 rather than SE = 0.98. We have now fixed this. In addition we agree with the reviewer that this method of quality control (i.e. empirically demonstrating that population stratification does not explain the bulk of the inflation in our GWAS test statistics) was likely unnecessary. However, due to the link between such social phenotypes and geography we felt it best to empirically demonstrate our claim that population stratification was unlikely to be a major contributing factor to these results.

Comment: “Why say "SEP" instead of the well-known "SES" or "social class?" Changing terminology and abbreviations to be confusing. At least once I had to go back and remind myself of what SEP means.”

Response: We agree with Reviewer #2 that we need to be consistent with the phenotype descriptions. We have now removed reference to socioeconomic status and included only socioeconomic position which is used extensively in the sociological and social-epidemiological literature.^{28, 29, 30}

Comment: On p. 7, it says ">10 putative third degree relatives from the kinship table." What does this mean?

Response: We have now made this clearer. The revised sentence now reads

“Finally, individuals with more than 10 putative third degree relatives (identified by Bycroft et al.¹⁵ by estimating the kinship coefficients for all pairs of samples using the software KING³¹) were also removed.”

Comment: I suggest either using an equation to give the definition of LD Score on p. 11 or not giving it at all. The present verbal definition is hard to understand.

Response: We agree with Reviewer #2. The verbal description has now been removed.

Comment: In the heritability-partitioning analyses with stratified LD Score regression (s-LDSC), the authors use the original annotations supplied by Finucane et al. (2015). I recommend the use of the Gazal et al. (2017) annotations, which include the original s-LDSC annotations as a subset. The additional annotations refer to properties such as LD itself, MAF, allele age, and so forth. These are important to include because heritability per SNP varying by LD is a potentially major source of confounding in LD Score regression. I make this a minor point rather than a major one because my experience suggests that the rank orders of enrichments and so forth are similar with or without the Gazal et al. annotations; with the new annotations, the enrichment estimates tend to become closer to one.

Response: We agree with Reviewer #2 that the continuous annotations should have been included in the manuscript. These have now been added and the results can be found in **Figure 5A, and B**, as well as in **Supplementary Tables 14, and 15**.

Comment: The cell-type data used by Skene et al. (2018),³² which can be employed with MAGMA or s-LDSC, has been used in studies of three phenotypes to my knowledge: schizophrenia (Skene et al., 2018), neuroticism (Nagel et al., 2018), and intelligence (Savage et al., 2018) (which shows a

strong genetic correlation with income). All three of these papers report significant or just-short-of-significant enrichment of medium spiny neurons and serotonergic neurons. It happens that the paper under review reports the same. I cannot point to any specific artefact in the data source or analysis, but I nevertheless find these results hard to accept. I suggest relegating these results to the supplement or inserting a caveat regarding the oddity of these a priori unlikely results being found regardless of the phenotype.

Response: We disagree with Reviewer #2 on their thoughts regarding this enrichment. The reviewer is correct that these analyses indicate gene expression that is unique to the medium spiny neurons and serotonergic neurons as being associated with a GWAS on intelligence+education as well as with neuroticism. However, as we have shown in our manuscript, there are strong genetic correlations between income with intelligence ($r_g = 0.69$, $SE=0.02$, $P>10\times 10^{-200}$), education ($r_g = 0.78$, $SE=0.01$, $P>10\times 10^{-200}$), and neuroticism ($r_g = -0.41$, $SE=0.08$, $P=6.97\times 10^{-8}$). Whereas these genetic correlations do not necessarily mean that the same gene sets will be associated with all these traits, the finding that they are is consistent with such a genetic correlation. Furthermore, in the original manuscript where these gene sets were derived,³² height was included as a “Non-brain comparator” to assess if the results of an enrichment of the serotonergic and medium spiny neurons was driven by an artefact as suggested by the reviewer. What was found by Skene et al. (2018)³² was that for height (which shows only a very small genetic correlation with intelligence, $r_g = 0.12$ ³³) no-enrichment of the serotonergic and medium spiny neurons was found (Supplementary Figure 9 of Skene et al. 2018³²).

Comment: *The authors use the original ten tissue annotations provided by Finacane et al. (2015) in their s-LDSC analyses. Each of these annotations, however, is formed by taking a union of SNPs associated with assayed histone marks across a variety of cell types and developmental stages. The SNPs that bear one of these annotations may be in fact quite heterogeneous. Therefore I suggest using the new and more specific tissue annotations provided by Finucane et al. (2018).*

Also, even if the authors keep their analyses with the 2015 annotations, I object to this paragraph on p. 26: "Whereas the tissue of the central nervous system showed the highest level of enrichment, the adrenal/pancreas, skeletal muscle, cardiovascular, and immune/hematopoietic tissues all showed significant enrichment. The finding that the regions of the genome undergoing purifying selection, as well as tissue types from multiple biological systems are enriched in their associations with income, is consistent with the notion that, whereas intelligence differences might make some contributions to differences in income, a range of other partly-heritable phenotypes also likely to contribute." But the enrichment of other tissues may simply mean that SNPs associated with histone marks in one tissue

may also be associated with such marks in another tissue. The authors need to look at the taus, that is, the partial regression coefficients of annotations.

Response: Reviewer #2 is correct. We have now conducted this analysis using the annotations from the updated Finucane et al. (2018)³⁴ (link to method <https://github.com/bulik/ldsc/wiki/Cell-type-specific-analyses>) paper. These results can be found in **Figure 5C-F** and **Supplementary Tables 16, 17, 18, and 19**. In addition, as the results have changed with the use of these updated data sets, we have removed the section incorrectly stating

"Whereas the tissue of the central nervous system showed the highest level of enrichment, the adrenal/pancreas, skeletal muscle, cardiovascular, and immune/hematopoietic tissues all showed significant enrichment. The finding that the regions of the genome undergoing purifying selection, as well as tissue types from multiple biological systems are enriched in their associations with income, is consistent with the notion that, whereas intelligence differences might make some contributions to differences in income, a range of other partly-heritable phenotypes also likely to contribute."

Comment: *The authors run s-LDSC on their income results that have been "enhanced" by MTAG. I regard the main uses of MTAG as identifying more genome-wide significant hits (and then performing downstream analyses of those hits) and obtaining a higher prediction R^2 with the PGS. I am skeptical of whole-genome analyses such as s-LDSC, where contamination of one phenotype by the other might be more of a concern. There needs to be more study of this issue. I approve of the authors relegating the s-LDSC results of MTAG-enhanced income to the supplement, but perhaps even there the authors should issue a caveat.*

Response: We share the concerns of the reviewer and indeed there are instances where the application of MTAG appears to have led to the creation of phenotypes that differed considerably from the target phenotype that they were supposed to enhance.^{22, 23} We have now amended the results section where we now include the sentence:

"It should be noted that in some instances application of MTAG appears to have led to the creation of a phenotype that differed considerably from the target phenotype that MTAG was supposed to enhance.^{22, 23} In order to explore this effect in our own data we derived the maxFDR statistic and performed genetic correlations with a data set composed only of participants who contributed data describing their level of household income. The maxFDR

gives the maximum FDR for the corresponding phenotype (in this case income), so if there are effects in our MTAG derived phenotype that are truly associated with education, but not income, there would also be an inflation of the FDR. The maxFDR derived was 0.003, over an order of magnitude lower than the commonly accepted standard of false discovery and comparable with those reported previously,^{33,35} indicating that the data set was capturing variance associated with income.

By comparing the genetic correlation of our MTAG derived phenotype with an income data set derived using only measures of income,¹⁴ we can determine if the genetic contributions of our MTAG phenotype differ from a phenotype derived using only measures of income.¹⁴ We find that the genetic correlation between our MTAG-income phenotype and a previous GWAS on income¹⁴ was $r_g=0.97$ (SE=0.024), with a genetic correlation of $r_g=0.94$ (SE=0.004) with educational attainment. This indicates that the polygenic signal in the MTAG-income analysis is virtually identical to that found in previous GWAS of income, but also that it captures more of the variance that is shared between income and education.”

This can be found on page 27.

Comment: *On p. 26, the authors say: "Medium spiny neurons have previously been linked to schizophrenia⁴² as well as to education.¹⁰" Reference 10 is Okbay et al. (2016), which did not use this method/dataset in the analysis of enriched tissues and cell types. So this thought is either wrong or incompletely expressed. In any event, if the authors follow one of my previous suggestions, these findings will be relegated to the supplement.*

Response: The reviewer is correct. We have now removed reference of medium spiny neurons being previously linked with education.

Reviewer: *On p. 27, the top paragraph is about evolutionarily conserved regions. But as far as I know, all phenotypes studied in GWAS show this enrichment. There is nothing special about income in this regard. It is thus a bit odd to make this a prominent finding for discussion.*

Response: We agree with the reviewer and have removed this paragraph.

Reviewer: *On p. 27, the authors say: "Also consistent with the action of a mutation-selection balance was the observed correlation of 0.42 ($P=2.2 \times 10^{-4}$) between minor allele frequency and the effect*

size for the SNPs found in the independent genomic loci – this indicated that variants with a lower MAF have a greater association with income." This relationship is almost certainly real, but its magnitude will be exaggerated if there is no correction for statistical power. I suggest not ascertaining for significance and plotting the data in the manner of Yang et al. (2015)'s Fig. 4C and D. The method used to calculate the statistical significance of the trend in that paper is now admitted by the authors to be incorrect. If desired, statistical significance is likely to be calculated accurately by the block jackknife.

Response: The reviewer is correct. We have now removed this paragraph.

References

1. Hill WD, *et al.* Genetic contributions to two special factors of neuroticism are associated with affluence, higher intelligence, better health, and longer life. *Molecular Psychiatry*, (2019).
2. Bowden J, Davey Smith G, Burgess S. Mendelian randomization with invalid instruments: effect estimation and bias detection through Egger regression. *International Journal of Epidemiology* **44**, 512-525 (2015).
3. Sanderson E, Davey Smith G, Windmeijer F, Bowden J. An examination of multivariable Mendelian randomization in the single-sample and two-sample summary data settings. *International Journal of Epidemiology*, (2018).
4. Burgess S, Thompson SG. Multivariable Mendelian Randomization: The Use of Pleiotropic Genetic Variants to Estimate Causal Effects. *American Journal of Epidemiology* **181**, 251-260 (2015).
5. Koellinger PD, Harden KP. Using nature to understand nurture. *Science* **359**, 386-387 (2018).
6. Kong A, *et al.* The nature of nurture: Effects of parental genotypes. *Science* **359**, 424-428 (2018).
7. Brumpton B, *et al.* Within-family studies for Mendelian randomization: avoiding dynastic, assortative mating, and population stratification biases. *bioRxiv*, 602516 (2019).
8. Leake J. Scientists find 24 'golden' genes that help you get rich. *The Times*, Retrieved from <https://www.thetimes.co.uk> (2019).
9. Marmot MG, Shipley MJ, Rose G. INEQUALITIES IN DEATH—SPECIFIC EXPLANATIONS OF A GENERAL PATTERN? *The Lancet* **323**, 1003-1006 (1984).
10. Anderson EL, *et al.* Education, intelligence and Alzheimer's disease: Evidence from a multivariable two-sample Mendelian randomization study. *bioRxiv*, (2018).
11. Sanderson E, Macdonald-Wallis C, Davey Smith G. Negative control exposure studies in the presence of measurement error: implications for attempted effect estimate calibration. *International Journal of Epidemiology* **47**, 587-596 (2018).
12. Hemani G, Tilling K, Davey Smith G. Orienting the causal relationship between imprecisely measured traits using GWAS summary data. *PLOS Genetics* **13**, e1007081 (2017).

13. Cesarini D, Lindqvist E, Östling R, Wallace B. Wealth, Health, and Child Development: Evidence from Administrative Data on Swedish Lottery Players *. *The Quarterly Journal of Economics* **131**, 687-738 (2016).
14. Hill WD, *et al.* Molecular genetic contributions to social deprivation and household income in UK Biobank. *Current Biology* **26**, 3083-3089 (2016).
15. Bycroft C, *et al.* The UK Biobank resource with deep phenotyping and genomic data. *Nature* **562**, 203-209 (2018).
16. Davies G, *et al.* Genome-wide association study of cognitive functions and educational attainment in UK Biobank (N=112 151). *Molecular Psychiatry* **21**, 758 (2016).
17. Okbay A, *et al.* Genome-wide association study identifies 74 loci associated with educational attainment. *Nature* **533**, 539 (2016).
18. Bulik-Sullivan B, *et al.* An atlas of genetic correlations across human diseases and traits. *Nature Genetics* **47**, 1236 (2015).
19. O'Connor LJ, Price AL. Distinguishing genetic correlation from causation across 52 diseases and complex traits. *Nature Genetics* **50**, 1728-1734 (2018).
20. Howey R, Shin S-Y, Relton C, Smith GD, Cordell HJ. Bayesian network analysis complements Mendelian randomization approaches for exploratory analysis of causal relationships in complex data. *bioRxiv*, 639864 (2019).
21. Kraja AT, *et al.* Associations of Mitochondrial and Nuclear Mitochondrial Variants and Genes with Seven Metabolic Traits. *The American Journal of Human Genetics Decision*, (Forthcoming).
22. Hill WD. Comment on 'Large-Scale Cognitive GWAS Meta-Analysis Reveals Tissue-Specific Neural Expression and Potential Nootropic Drug Targets' by Lam *et al.* *Twin Research and Human Genetics* **21**, 84-88 (2018).
23. Hill WD. A Further Comment on 'Large-Scale Cognitive GWAS Meta-Analysis Reveals Tissue-Specific Neural Expression and Potential Nootropic Drug Targets' by Lam *et al.* *Twin Research and Human Genetics*, 1-8 (2018).
24. Kang HJ, *et al.* Spatio-temporal transcriptome of the human brain. *Nature* **478**, 483 (2011).
25. Byrne EM, *et al.* Conditional GWAS analysis identifies putative disorder-specific SNPs for psychiatric disorders. *bioRxiv*, 592899 (2019).

26. de Leeuw CA, Mooij JM, Heskes T, Posthuma D. MAGMA: Generalized Gene-Set Analysis of GWAS Data. *PLoS Comp Biol* **11**, (2015).
27. Bulik-Sullivan BK, *et al.* LD Score regression distinguishes confounding from polygenicity in genome-wide association studies. *Nature Genetics* **47**, 291 (2015).
28. Krieger N, Williams DR, Moss NE. Measuring Social Class in US Public Health Research: Concepts, Methodologies, and Guidelines. *Annual Review of Public Health* **18**, 341-378 (1997).
29. Galobardes B, Shaw M, Lawlor DA, Lynch JW, Davey Smith G. Indicators of socioeconomic position (part 1). *Journal of Epidemiology and Community Health* **60**, 7-12 (2006).
30. Galobardes B, Shaw M, Lawlor DA, Lynch JW, Davey Smith G. Indicators of socioeconomic position (part 2). *Journal of Epidemiology and Community Health* **60**, 95-101 (2006).
31. Manichaikul A, Mychaleckyj JC, Rich SS, Daly K, Sale M, Chen W-M. Robust relationship inference in genome-wide association studies. *Bioinformatics* **26**, 2867-2873 (2010).
32. Skene NG, *et al.* Genetic identification of brain cell types underlying schizophrenia. *Nature Genetics* **50**, 825-833 (2018).
33. Hill W, *et al.* A combined analysis of genetically correlated traits identifies 187 loci and a role for neurogenesis and myelination in intelligence. *Molecular psychiatry*, 1 (2018).
34. Finucane HK, *et al.* Heritability enrichment of specifically expressed genes identifies disease-relevant tissues and cell types. *Nature Genetics* **50**, 621-629 (2018).
35. Turley P, *et al.* Multi-trait analysis of genome-wide association summary statistics using MTAG. *Nature Genetics*, (2018).

Reviewers' Comments:

Reviewer #1:

Remarks to the Author:

Report for MS #NC-2019-2 GWAS of income

I liked the original submission of the paper, and I like the revised version even more. I especially like the addition of the FAQs, which should help defuse some of the potential controversy and misinterpretations about the paper (and which the authors can send to journalists who inquire about the paper).

Here are my comments on the revised paper:

1. Related to my first major comment on the original submission: I like that the authors have added qualifiers (e.g., "if the assumptions of MR in this instance are met"), but in my opinion, they should go further in stating clearly that the assumptions of MR (or its various generalizations) are unlikely to be met in the present context. In the Discussion, the authors highlight dynastic effects as a likely violation, but horizontal pleiotropy that violates the MR assumptions is surely widespread and also worth highlighting.

Also, the sentence "...since there was no evidence of directional pleiotropy, the overall causal estimate based on all of the genetic variants is unlikely to be biased" is not correct (e.g., the InSIDE assumption may be violated).

2. Related to my fifth major comment on the original submission: although the exposition is clearer than in the previous draft, I think the paper could be even clearer that it is treating the 1-5 categories of income as a continuous variable. I also still think the paper would be stronger if it included a robustness analysis with an ordinal regression. (The fact that results for educational attainment are similar when the variable is treated as binary versus continuous does not tell us that the same is true for income.)

3. While the FAQs are generally excellent, I would quibble with this paragraph: "We found that variation across all the SNPs in the DNA from this sample accounted for 7.4% of the variation in household income...Therefore, as we noted above, the large majority of people's differences in household income are likely to be environmental in origin, according to our results." This paragraph seems to overstate the case. Don't twin studies estimate the heritability of SEP to be roughly 40%? In that case, the non-genetic component (60%) isn't a "large majority." Also, while dominance and epistasis may or may not comprise much of the overall heritability of SEP, it seems like they should be somehow included in the list in the paragraph.

4. There are many typos throughout the paper, so the authors should go through it carefully. Here are some of the ones I caught:

i. Abstract: "the genetic variants associated with income are related to better mental health than those linked to educational attainment"

ii. "an LD regression"

iii. "LDSC score regression"

iv. "whilst a general factor of neuroticism shows a negative genetic correlation with household income...whereas the two special factors of neuroticism"

v. "We were able to predict between up to 2.00% of variance in income"

vi. "would be partially due to the effects we observe may be due to the of the parents' genotype"

vii. "~30%of"

Reviewer #3:

Remarks to the Author:

Having read Reviewer 2's comments, I find the authors' responses satisfactory in all cases except

for the following:

1. Comment: On p. 11, is "A minor allele frequency cut-off of >0.1 " a typo?

Response: "This is not a typo and these are the standard MAF and INFO score cut offs used for LDSC regression. This sentence has been made clearer –we now state that "SNPs were included if they had a minor allele frequency of > 0.1 and an imputation quality score of > 0.9 ."

The authors must be confused here. A MAF threshold of 0.1 is absurdly high. The default in LD score regression is 0.01.

2. Comment: "On p. 24, why is the SE of the LDSC intercept so large? This is baffling. And why was this analysis even done? If I was confident that neither trait's GWAS was subject to much confounding, I would use MTAG without too much concern."

Response: "We thank the reviewer for their attention to detail here. This figure was misreported and should have stated $SE = 0.0098$ rather than $SE = 0.98$. We have now fixed this. In addition we agree with the reviewer that this method of quality control (i.e. empirically demonstrating that population stratification does not explain the bulk of the inflation in our GWAS test statistics) was likely unnecessary. However, due to the link between such social phenotypes and geography we felt it best to empirically demonstrate our claim that population stratification was unlikely to be a major contributing factor to these results."

I agree with the previous R2 that it does not make sense to do this analysis, for the additional reason that MTAG adjusts the standard errors and P-values by the square-root of the LD score regression intercept. So the LD score intercept obtained by using these results is pretty much meaningless. Similarly, the heritability estimate is biased. If the authors want to look at the amount the stratification in the MTAG results or estimate the heritability, they should switch off that option in MTAG and use results that have not been adjusted for the intercept.

Additional comments

3. Page 7, line 175: Are these really the only quality control filters applied? Have the authors checked any QC plots? A MAF threshold of 0.0005 is extremely small, even for a sample as large as UKB, and I have never seen someone apply an imputation quality threshold as low as 0.1. The standard is to adopt a threshold between 0.4-0.9. This is a problem especially since both the MAF and imputation accuracy thresholds are low. I suggest the authors consult the Pistis et al. (2015) paper for a more appropriate imputation accuracy threshold for rare variants.

4. Have the authors adjusted the standard errors and P-values for lambdaGC or the LD score regression intercept to correct for residual stratification?

5. For a phenotype like income, age-sex interaction seems to me like an important thing to control for in the GWAS. Similarly, non-linear age effects could capture some variance. Why have the authors elected not to control for these covariates?

6. The description of independent SNPs, independent loci and lead SNPs on page 8 is very confusing. Please make that clearer.

7. Page 14, line 357: More detail is required on the new GWAS of intelligence that the authors use for the Mendelian Randomization analyses. It is not clear which publicly available data were meta-analyzed with the newly conducted GWAS. I also could not find any information about the newly conducted INTERVAL GWAS anywhere in the paper.

8. In the PRS analyses, why do the authors first residualize income on the covariates, and then use these residuals as the outcome variable? This would be fine if they residualized the polygenic scores on the covariates (or just the PCs) as well but this does not seem to be the case. The

current setting is not sufficient to control for population stratification. Moreover, why do the authors not control for age-sex interaction and age2 here either?

9. Page 18, line 454: This is a good point. One way to look at how important genetic nurture is for income is of course to conduct within-family association analyses, which are missing from the paper. This would also be a good way to deal with concerns about residual population stratification, which is a valid concern for a phenotype like income.

10. Given the high genetic correlation between income and educational attainment, it would be nice to see how well scores based on the MTAG output for educational attainment predict income.

References

Pistis, G., Porcu, E., Vrieze, S. I., Sidore, C., Steri, M., Danjou, F., ... Sanna, S. (2015). Rare variant genotype imputation with thousands of study-specific whole-genome sequences: implications for cost-effective study designs. *European Journal of Human Genetics*, 23(7), 975–983. <https://doi.org/doi:10.1038/ejhg.2014.216>

Reviewer #1

Comment: “I liked the original submission of the paper, and I like the revised version even more. I especially like the addition of the FAQs, which should help defuse some of the potential controversy and misinterpretations about the paper (and which the authors can send to journalists who inquire about the paper).”

Response: We are grateful for the reviewer’s appraisal of our work and thank them for their suggestion to include an FAQ.

Comment: “Related to my first major comment on the original submission: I like that the authors have added qualifiers (e.g., “if the assumptions of MR in this instance are met”), but in my opinion, they should go further in stating clearly that the assumptions of MR (or its various generalizations) are unlikely to be met in the present context. In the Discussion, the authors highlight dynastic effects as a likely violation, but horizontal pleiotropy that violates the MR assumptions is surely widespread and also worth highlighting.

Also, the sentence “...since there was no evidence of directional pleiotropy, the overall causal estimate based on all of the genetic variants is unlikely to be biased” is not correct (e.g., the InSIDE assumption may be violated).”

Response: We agree that the genetic variants identified in the intelligence GWAS are likely to have pleiotropic effects. However, this is not sufficient to cause bias in Mendelian randomization analyses. The genetic variants must have horizontally pleiotropic effects – that is effects on the outcome (income) that are not mediated via intelligence. It is currently unclear what these phenotypes are. We have added the following to the discussion.

“Furthermore, genetic variants associated with intelligence are likely to have pleiotropic effects.¹ However, to break the assumptions of Mendelian randomization it is not sufficient for the genetic variants to have pleiotropic effects.² The genetic variants we use as instruments must have horizontally pleiotropic effects mediated via mechanisms other than intelligence. If the genetic variants have vertically pleiotropic effects, e.g. SNP->neuron->intelligence->education->health, then our Mendelian randomization estimates will not be biased. Equally, if the SNPs affect other phenotypes, but these phenotypes do not affect outcome, then these effects will not result in bias in the Mendelian randomization estimates. It is possible that the genetic variants identified in intelligence GWAS have horizontally

pleiotropic effects, however, it is unclear what mechanisms would mediate these effects. The genetic correlations between intelligence and personality traits are relatively low.³ The genetic variants identified in the intelligence GWAS are likely to also affect a range of cognitive ability related traits. However again, these pleiotropic effect via related phenotypes are unlikely to cause bias if the results are interpreted as a test of general cognitive function. It is possible to investigate potentially horizontal pleiotropic effects further using multivariable Mendelian randomization.⁴ If SNPs have been identified that explain sufficient independent variation in two or more two potential pathways, e.g. intelligence and education, it is possible to identify the direct effects of each exposure. Future research should use multivariable Mendelian randomization to investigate this further.”

In order to address the reviewer’s concerns within our manuscript we have made changes to the text. These changes are on page 23 under the heading of “Causal links with intelligence” and are highlighted in bold I the text below.

“**Should the assumptions of MR be met**, this indicates that greater intelligence causes a higher level of income. Sensitivity analyses revealed little evidence of directional pleiotropy which can bias MR **estimates (MR-Egger intercept=0.010, SE=0.007, P=0.189)** (Supplementary Table 20). The heterogeneity statistics indicate that the **estimated size** of the causal effect **of intelligence on income** varies across the SNPs (Supplementary Table 20). However, since there was **little** evidence of directional pleiotropy, the overall causal estimate based on all of the genetic variants is unlikely to be biased **if the MR-Egger assumptions hold (i.e. the InSIDE assumption).**”

Comment: “Related to my fifth major comment on the original submission: although the exposition is clearer than in the previous draft, I think the paper could be even clearer that it is treating the 1-5 categories of income as a continuous variable. I also still think the paper would be stronger if it included a robustness analysis with an ordinal regression. (The fact that results for educational attainment are similar when the variable is treated as binary versus continuous does not tell us that the same is true for income.)”

Response: We agree with the reviewer and have now modified the description of the phenotype to read as follows

“Phenotype description

A total of 332,050 participants had genotype data and data on their level of household income. Self-reported household income was collected using a 5 point scale corresponding to

the total household income before tax, 1 being less than £18,000, 2 being £18,000 - £29,999, 3 being £30,000 - £51,999, 4 being £52,000 – £100,000, and 5 being greater than £100,000. Participants were removed from the analysis if they answered “do not know” (n = 12,721), or “prefer not to answer” (n = 31,947). This left a total number of 286,301 participants (138,425 male) aged 39-73 years (mean = 56.5, SD = 8.0 years) with genotype data who had reported, between 1 and 5, their level of household income. **This 5 point scale was analysed by treating the categories of income as a continuous variable.**”

Comment: “While the FAQs are generally excellent, I would quibble with this paragraph: “We found that variation across all the SNPs in the DNA from this sample accounted for 7.4% of the variation in household income... Therefore, as we noted above, the large majority of people’s differences in household income are likely to be environmental in origin, according to our results.” This paragraph seems to overstate the case. Don’t twin studies estimate the heritability of SEP to be roughly 40%? In that case, the non-genetic component (60%) isn’t a “large majority.” also, while dominance and epistasis may or may not comprise much of the overall heritability of SEP, it seems like they should be somehow included in the list in the paragraph.”

Response: The reviewer is correct and although we did state that some of the 92.6% of variation not explained by our GWAS may be explained by genetic differences that we didn’t measure, this information could be presented in a clearer manner. The paragraph has now been revised to read as follows

“We found that variation across all the SNPs measured in the current study from the UK Biobank sample accounted for 7.4% of the variation in household income. This is a small amount and other studies that have used methods to capture less common genetic variation have found heritability estimates for income between 40-50%.⁵ Therefore, it is likely that our estimate, based only on common genetic variation, is an underestimation of the total genetic effect.

The variation in income that is not due to genetic effects is likely due to environmental factors, as well as errors of measurement (for example, people often make mistakes filling in questionnaires, including questionnaires about income, and this creates some “noise” in the measure). Therefore, as we noted above, a large proportion of people’s differences in household income is likely to be environmental in origin, according to our results.”

Comment: “There are many typos throughout the paper, so the authors should go through it carefully. Here are some of the ones I caught:

- i. Abstract: “the genetic variants associated with income are related to better mental health than those linked to educational attainment”
- ii. “an LD regression”
- iii. “LDSC score regression”
- iv. “whilst a general factor of neuroticism shows a negative genetic correlation with household income...whereas the two special factors of neuroticism”
- v. “We were able to predict between up to 2.00% of variance in income”
- vi. “would be partially due to the effects we observe may be due to the of the parents’ genotype”
- vii. “~30%of”

Response: We thank the reviewer for their attention to detail. We have made the corrections to the typos found by the reviewer and thoroughly checked the manuscript for any others that may be present. The typos highlighted by the reviewer now read as follows:

- i. the genetic variants associated with income are related to better mental health than the genetic variants associated with education
- ii. LD score regression
- iii. LDSC regression
- iv. whilst a general factor of neuroticism shows a negative genetic correlation with household income... the two special factors of neuroticism
- v. We were able to predict between up to 2% of the variance in income
- vi. would be partially due to the effects of the parents’ genotype on their parents’ intelligence which subsequently affects offspring income
- vii. ~30% of

Reviewer #3

Comment: “The authors must be confused here. A MAF threshold of 0.1 is absurdly high. The default in LD score regression is 0.01.”

Response: The reviewer is correct. This figure should be reported as the LDSC default which is 0.01 or 1%. This has now been amended in the manuscript.

Comment: “I agree with the previous R2 that it does not make sense to do this analysis, for the additional reason that MTAG adjusts the standard errors and P-values by the square-root of the LD score regression intercept. So the LD score intercept obtained by using these results is pretty much meaningless. Similarly, the heritability estimate is biased. If the authors want to look at the amount the stratification in the MTAG results or estimate the heritability, they should switch off that option in MTAG and use results that have not been adjusted for the intercept.”

Response: We thank the reviewer for their comment here. We have now removed the paragraph describing the heritability and LDSC regression intercept for the MTAG derived income phenotype.

Comment: “Page 7, line 175: Are these really the only quality control filters applied? Have the authors checked any QC plots? A MAF threshold of 0.0005 is extremely small, even for a sample as large as UKB, and I have never seen someone apply an imputation quality threshold as low as 0.1. The standard is to adopt a threshold between 0.4-0.9. This is a problem especially since both the MAF and imputation accuracy thresholds are low.”

Response: We applied the same quality control filters here that have been applied previously.^{6,7} We would also like to point out that the quality control filters mentioned by the reviewer are those that occurred after the association analysis. We have copied in the full description of the quality control performed below.

“Full details of the UK Biobank genotyping procedure have been made available.⁸ In brief, two custom genotyping arrays were used to genotype 49,950 participants (UK BiLEVE Axiom Array) and 438,427 participants (UK Biobank Axiom Array).^{8,9} Genotype data on 805,426 markers were available for 488,377 of the individuals in UK Biobank. Imputation to the Haplotype Reference Consortium (HRC) reference panel lead to 39,131,578 autosomal

SNPs being available for 487,442 participants.⁸ Allele frequency checks¹⁰ against the HRC¹¹ and 1000G¹² site lists were performed, and variants with minor allele frequencies (MAF) differing more than +/- 0.2 from the reference sets were removed.

Additional quality control steps were conducted and described previously.^{6,13} These included the removal of those with non-British ancestry based on self-report and a principal components analysis, as well as those with extreme scores based on heterozygosity and missingness. Individuals with neither XX or XY chromosomes, along with those individuals whose reported sex was inconsistent with genetically inferred sex, were also removed. Finally, individuals with more than 10 putative third degree relatives (identified by Bycroft et al.⁸ by estimating the kinship coefficients for all pairs of samples using the software KING¹⁴) were also removed. Following these exclusions, a sample of 408,095 individuals remained. Using GCTA-GREML on 131,790 reportedly-related participants,¹⁵ related individuals were removed based on a genetic relationship threshold of 0.025. Following this quality control, household income data, and genetic data, were available on 286,301 participants. Following association analysis, SNPs with a minor allele frequency < 0.0005, and an imputation quality score < 0.1 were removed. Finally, only bi-allelic SNPs were retained, resulting in 18,485,882 autosomal SNPs.”

Comment: “Have the authors adjusted the standard errors and P-values for lambdaGC or the LD score regression intercept to correct for residual stratification?”

Response: We did not adjust the standard errors and p values for residual stratification as the LDSC intercept was 1.04 indicating only a very small amount of residual stratification in our data. When we derived the ratio of $(LDSC\ intercept - 1) / (\text{mean}(\chi^2) - 1)$ describing the proportion of the inflation in the mean χ^2 due to factors other than polygenicity and found it to be 0.0794 indicating that over 92% of the inflation in the mean χ^2 statistic was due to polygenicity. On the basis of these results we decided not to adjust our results further.

Comment: “For a phenotype like income, age-sex interaction seems to me like an important thing to control for in the GWAS. Similarly, non-linear age effects could capture some variance. Why have the authors elected not to control for these covariates?”

Response: We would agree with the reviewer if we had individual level data however our phenotype is at the level of “household”. This “household” is based on address, and so

individuals who reside in the same army barracks, care homes, hospitals etc, would all count as living together. This makes an interpretation of what an age-sex interaction is measuring unclear, and so we elected not to include interaction terms in our analyses. Similarly, an age interaction was not included as the phenotype each participant had is a product of everyone, young and old, living in the same address.

Comment: “The description of independent SNPs, independent loci and lead SNPs on page 8 is very confusing. Please make that clearer.”

Response: We agree with the reviewer that this could be clearer. The revised text now reads as follows.

“Genomic risk loci were derived using the summary data from the data set of household income derived in UK Biobank, using Functional Mapping and Annotation of genetic associations (FUMA).¹⁶ First, independent significant SNPs were defined using a P-value cut off of genome-wide significant ($P < 5 \times 10^{-8}$), as well as being independent from each other ($r^2 < 0.6$) within a 1mb window. Second, SNPs that were in LD with any independent SNP ($r^2 \geq 0.6$) and within a 1mb window in addition to being in the HRC genomes reference panel with a MAF greater than 0.001, were included for further annotation. Third, lead SNPs were identified using the independent significant SNPs as defined above. Lead SNPs were a subset of the independent significant SNPs that were in LD with each other at $r^2 < 0.1$, with a 1mb window. Fourth, genomic risk loci were created by merging lead SNPs if they were closer than 250 kb apart. This means that a genomic risk locus could contain multiple independent significant SNPs and multiple lead SNPs. Finally, all SNPs in LD of $r^2 \geq 0.6$ with one of the independent significant SNPs formed the border, or edge, of the genomic risk loci.”

Comment: “Page 14, line 357: More detail is required on the new GWAS of intelligence that the authors use for the Mendelian Randomization analyses. It is not clear which publicly available data were meta-analyzed with the newly conducted GWAS. I also could not find any information about the newly conducted INTERVAL GWAS anywhere in the paper.”

Response: The information on how the independent groups were created for the Mendelian Randomisation analysis can be found in the Supplemental data section. This section contains the information of which publically available data set was used for analysis, the genotyping and imputation of the INTERVAL data, as well as how the intelligence phenotype was constructed and how the meta-analysis was conducted.

Comment: “In the PRS analyses, why do the authors first residualize income on the covariates, and then use these residuals as the outcome variable? This would be fine if they residualized the polygenic scores on the covariates (or just the PCs) as well but this does not seem to be the case. The current setting is not sufficient to control for population stratification. Moreover, why do the authors not control for age-sex interaction and age² here either?”

Response: The reviewer is correct that we residualized the income phenotype from Generation Scotland for the 20 principal components as well as age and sex. This is the same method we used to control for population stratification as performed in our original GWAS in the current paper and have performed in many other GWAS.^{6,7,17} For each of these GWAS we were able to perform LDSC regression to ensure that this method did remove the effects of population stratification and in each instance the intercept from the LDSC regression was close to zero indicating no, or very little, population stratification. By applying this same method to the income phenotype in Generation Scotland we are confident that residual stratification has been dealt with in our analysis.

Comment: “Page 18, line 454: This is a good point. One way to look at how important genetic nurture is for income is of course to conduct within-family association analyses, which are missing from the paper. This would also be a good way to deal with concerns about residual population stratification, which is a valid concern for a phenotype like income.”

Response: The reviewer is correct that within family analysis would be a worthy inclusion to our manuscript. However, as mentioned in a response the reviewer above the phenotype is at the level of the household and not at the individual. This means that if we were to perform a within family design then all members of the same family would receive the same phenotype score, meaning that there would be no variance within each family to analyse.

Comment: “Given the high genetic correlation between income and educational attainment, it would be nice to see how well scores based on the MTAG output for educational attainment predict income.”

Response: We agree with the reviewer and have now conducted these analyses. The results of which can be found on page 28 and are copied in below for clarity.

“Following a reviewer’s request we used the MTAG-derived education phenotype to predict household income in Generation Scotland. Using this phenotype the p-value cut off with the most predictive validity was 0.05 which predicted 2.18% of the variance of income.”

References

1. Pickrell, J.K. *et al.* Detection and interpretation of shared genetic influences on 42 human traits. *Nature Genetics* **48**, 709 (2016).
2. Hemani, G., Bowden, J. & Davey Smith, G. Evaluating the potential role of pleiotropy in Mendelian randomization studies. *Human Molecular Genetics* **27**, R195-R208 (2018).
3. Trampush, J.W. *et al.* GWAS meta-analysis reveals novel loci and genetic correlates for general cognitive function: a report from the COGENT consortium. *Molecular Psychiatry* **22**, 336 (2017).
4. Sanderson, E., Davey Smith, G., Windmeijer, F. & Bowden, J. An examination of multivariable Mendelian randomization in the single-sample and two-sample summary data settings. *International Journal of Epidemiology* (2018).
5. Hyttinen, A., Ilmakunnas, P., Johansson, E. & Toivanen, O. Heritability of lifetime earnings. *The Journal of Economic Inequality* **17**, 319-335 (2019).
6. Luciano, M. *et al.* Association analysis in over 329,000 individuals identifies 116 independent variants influencing neuroticism. *Nature Genetics* **50**, 6-11 (2018).
7. Hill, W.D. *et al.* Genetic contributions to two special factors of neuroticism are associated with affluence, higher intelligence, better health, and longer life. *Molecular Psychiatry* (2019).
8. Bycroft, C. *et al.* The UK Biobank resource with deep phenotyping and genomic data. *Nature* **562**, 203-209 (2018).
9. Wain, L.V. *et al.* Genome-wide association analyses for lung function and chronic obstructive pulmonary disease identify new loci and potential druggable targets. *Nature genetics* **49**, 416 (2017).
10. Winkler, T.W. *et al.* Quality control and conduct of genome-wide association meta-analyses. *Nature protocols* **9**, 1192 (2014).
11. Haplotype Reference, C. A reference panel of 64,976 haplotypes for genotype imputation. *Nature genetics* **48**, 1279-1283 (2016).
12. Genomes Project, C. An integrated map of genetic variation from 1,092 human genomes. *Nature* **491**, 56 (2012).
13. Hill, W. *et al.* A combined analysis of genetically correlated traits identifies 187 loci and a role for neurogenesis and myelination in intelligence. *Molecular psychiatry*, 1 (2018).
14. Manichaikul, A. *et al.* Robust relationship inference in genome-wide association studies. *Bioinformatics* **26**, 2867-2873 (2010).
15. Yang, J., Lee, S.H., Goddard, M.E. & Visscher, P.M. GCTA: a tool for genome-wide complex trait analysis. *Am J Hum Genet* **88**, 76-82 (2011).
16. Watanabe, K., Taskesen, E., Bochoven, A. & Posthuma, D. Functional mapping and annotation of genetic associations with FUMA. *Nature communications* **8**, 1826 (2017).
17. Davies, G. *et al.* Study of 300,486 individuals identifies 148 independent genetic loci influencing general cognitive function. *Nature Communications* **9**, 2098 (2018).

Reviewers' Comments:

Reviewer #1:

Remarks to the Author:

The authors have been mostly responsive to my earlier comments, and I believe the paper is improved.

Here are my comments on the most recent paper:

1. The summary statistics from the new GWAS of intelligence using the INTERVAL data should be made publicly available but are not currently listed under "Data availability."
2. I agree with Reviewer #3's comment that "The description of independent SNPs, independent loci and lead SNPs on page 8 is very confusing. Please make that clearer." I think the revised text continues to be sufficiently clear for someone to replicate the analysis. For example, take the sentence "First, independent significant SNPs were defined using a P-value cut off of genome-wide significant ($P < 5 \times 10^{-8}$), as well as being independent from each other ($r^2 < 0.6$) within a 1mb window." This sentence is ambiguous about the algorithm: if two (or more) genome-wide-significant SNPs are not correlated with $r^2 \geq 0.6$, which is dropped? And in what order? Similarly for the definition of lead SNPs.
3. In the "Supplemental data" file, what is "Deriving independent groups for Mendelian randomization"? Is there a section missing?

Reviewer #3:

Remarks to the Author:

I thank the authors for addressing all my concerns (including the ones missing the point that this is a GWAS on family income, not individual income, sorry!). I have no further comments and recommend the manuscript gets published without further revisions.

Reviewer #1

Comment: The summary statistics from the new GWAS of intelligence using the INTERVAL data should be made publicly available but are not currently listed under “Data availability.”

Response: These data will be made available.

Comment: I agree with Reviewer #3’s comment that “The description of independent SNPs, independent loci and lead SNPs on page 8 is very confusing. Please make that clearer.” I think the revised text continues to be sufficiently clear for someone to replicate the analysis. For example, take the sentence “First, independent significant SNPs were defined using a P-value cut off of genome-wide significant ($P < 5 \times 10^{-8}$), as well as being independent from each other ($r^2 < 0.6$) within a 1mb window.” This sentence is ambiguous about the algorithm: if two (or more) genome-wide-significant SNPs are not correlated with $r^2 \geq 0.6$, which is dropped? And in what order? Similarly for the definition of lead SNPs.

Response: We disagree that the text is too unclear for other groups to replicate the method used. FUMA is publically available and to replicate our analysis another group would use FUMA with the settings specified. However, we agree with the reviewer that should a group wish to replicate our analysis with their own tools then this section should be able to stand without reference to FUMA. We have now clarified the text to read as follows

“First, clumping was performed on all SNPs to identify independent significant SNPs. Independent significant SNPs were defined using a P-value cut off of genome-wide significant ($P < 5 \times 10^{-8}$), as well as being independent from each other ($r^2 < 0.6$) within a 1mb window. Second, SNPs that were in LD with any independent SNP ($r^2 \geq 0.6$) and within a 1mb window in addition to being in the HRC genomes reference panel with a MAF greater than 0.001, were included for further annotation. Third, lead SNPs were identified using the independent significant SNPs as defined above. Lead SNPs were identified by clumping independent significant SNPs that were in LD with each other at $r^2 < 0.1$, with a 1mb window. Fourth, genomic risk loci were created by merging lead SNPs if they were closer than 250 kb apart. This means that a genomic risk locus could contain multiple independent significant SNPs and multiple lead SNPs. Finally, all SNPs in LD of $r^2 \geq 0.6$ with one of the independent significant SNPs formed the border, or edge, of the genomic risk loci.”

Comment: In the “Supplemental data” file, what is “Deriving independent groups for Mendelian randomization”? Is there a section missing?

Response: The title “Deriving independent groups for Mendelian randomisation” pertains to using the publically available data on intelligence and combining it with the INTERVAL data on intelligence to make an independent group to perform Mendelian randomisation. No section is missing but in order to reduce confusion we have removed this header.

Reviewer #3

Comment: I thank the authors for addressing all my concerns (including the ones missing the point that this is a GWAS on family income, not individual income, sorry!). I have no further comments and recommend the manuscript gets published without further revisions.

Response: We thank the reviewer for their comments which have strengthened the manuscript.